# Trust Region Policy Optimization for Functional Linear Policies

## Abstract

Reinforcement Learning (RL) tasks where the states are given by spatial or temporal measurements often lead to high-dimensional state spaces, making function approximation difficult and unstable. We adapt the classic RL framework to allow the direct use of the inherent functional state, which can be estimated from the discrete measurements. We propose a suitable family of policies based on functional linear models, allowing us to take actions conditionally on functional states. Moreover, we extend Trust Region Policy Optimization (TRPO) to improve such policies and address the challenge of operator inversion in infinite-dimensional spaces using techniques from Functional Data Analysis (FDA). Furthermore, we implement Proximal Policy Optimization (PPO) for these policies. In experiments on three linear PDE control tasks, functional policies yield more stable training and achieve better performance than multilayer perceptron policies, highlighting the benefits of functional representations in RL.

## Introduction

Reinforcement Learning (RL) can be extended beyond a finite-dimensional setting to handle state measurements, which are functions in the infinite-dimensional space of square-integrable functions. Despite this added complexity, the core RL framework remains intact. With appropriately defined policies, standard policy gradient methods are still applicable—but their implementation requires inverting a linear operator in an infinite-dimensional space. We tackle this challenge by leveraging techniques from Functional Data Analysis (FDA).

From playing video games (Lample & Chaplot, 2017) to smart grid control (Arwa & Folly, 2020) and even solving control problems (Recht, 2019; Zhang et al., 2024), the RL paradigm has allowed to tackle an important number of real-world problems. Markov Decision Processes (MDP) are the backbone of most RL methods, which rely on a Markovian hypothesis: future states and future rewards depend on past states and actions only through the last state and action; as a direct consequence, the policy must choose an action using only the current state. To use temporal information, a popular workaround is to augment the state space, for example, by including the last 4 frames of an Atari game (Mnih et al., 2015) or using the 24 hourly measures of electric load and electricity prices to dynamically manage electricity production (Ji et al., 2019). Another type of relevant state space is of spatial character, consisting of measurements of the same quantity across a fine grid; for example, in the *heat invader* control problem, an action must be chosen depending on heat measures in a $64 \times 64$ spatial grid (Farahmand et al., 2017).

Spatial or temporal states are challenging due to their high-dimensional nature. In the literature, these problems have mainly been approached using Deep Reinforcement Learning (DRL) by using value based methods with approximation (Mnih et al., 2015; Ji et al., 2019; Farahmand et al., 2017). In contrast, we develop policy based methods, approaching these high-dimensional problems from the FDA perspective, which is a popular approach to analyze data that are curves (Ramsay & Silverman, 2005) dealing with the infinite-dimensionality by using dimension reduction techniques such as *Functional Principal Component Analysis* (FPCA).

Adopting the framework introduced by Hernandez-Lerma (2001), we propose an MDP setting where the state space is a functional space, we define linear functional policies, and then adapt Trust Region Policy Optimization (TRPO) (Schulman et al., 2015a) to improve such policies. We show that the main theoretical results from TRPO generalize well into this setting, but the practical implementation is challenging because a linear-inverse problem must be solved in an infinite-dimensional space. We tackle these issues by exploring different FDA techniques: finite basis projection, FPCA (Wang et al., 2016) and a resolvent approach (Martini et al., 2022).

**Related work:** This functional setting has received increasing attention. Notably, functional states naturally arise in control problems where system evolution is governed by Partial Differential Equations (PDE). Farahmand et al. (2016) proposed an RL approach for PDE control using a value-based method: regularized fitted Q-iteration, adapting Q-learning for handling infinite dimensionality via a reproducing kernel Hilbert space representation. Later, Pan et al. (2018), addressed control problems with function-valued actions by exploiting spatial regularity in the action space through "action descriptors", reducing the infinite dimensional problem to a finite dimensional one, and then proposing a deterministic policy gradient algorithm in that context to improve actor and critic convolutional networks. At last, this infinite-dimensional setting may arise from "large-scale RL", where an "infinite" amount of agents must learn how to act, Zhang & Li (2025) tackle this setting by proposing an adapted RKHS approach.

An alternative to using RKHS or CNNs is to use a finite functional basis (Fourier, splines, etc.) expansion for function value approximation. This was one of the earliest forms of function approximation techniques for continuous multivariate states; early examples include the work of Schweitzer & Seidmann (1985) or Bertsekas & Tsitsiklis (1996, Chapter 3, Section 1), approximating the state or the state-action value functions as a linear combination of non-linear features of the state. For instance, Konidaris et al. (2011) approximates the state value function as $\bar{V}(s) \simeq \sum_{k=1}^{m} \omega_k \phi_k(s)$, with learnable weights and $\{\phi_k\}_{k=1}^{m}$ a multivariate Fourier basis. More recently, this approach was adapted to allow for non-linear function approximation with the work of Brellmann et al. (2023)) or Li & Pathak (2021). Both works propose parameterizing the Q-function with a neural network using a multivariate Fourier transform for the state space, which was then fed into an MLP to approximate the state-value function. The former work uses a fixed multivariate Fourier transform, while the latter incorporates a learned multivariate Fourier transform. It is worth stressing that in any case, the state is a multivariate scalar vector. Additionally, this functional setting is also present in image-based RL; for instance, Li & Pathak (2021) proposes a Q-learning method, incorporating a learnable Fourier embedding into the Q-network, improving sample efficiency for image-RL problems.

Another relevant modern approach to functional problems are neural operators (NO). These learn mappings between infinite-dimensional function spaces, which naturally have been used for PDE modelling. For example, Lu et al. (2021), relying on the universal approximation theorem for operators, propose a deep operator network (DeepONet), which successfully allowed simulating various dynamical systems and partial differential equations. Similarly, Li et al. (2021), also propose a NO by parametrizing a kernel on Fourier space, learning resolution-invariant PDE solutions. These promising NO approaches have been used for PDE control using RL approaches. For instance, Hu et al. (2025), use DeepONet as a feature extractor, which is then used for PDE control using a soft actor-critic RL approach. Nevertheless, NO approaches typically require pretraining on PDE data and result in black-box neural-networks.

**Main contributions:** In contrast with existing methods from the literature, we do not use an RKHS, NO approaches nor function value approximation; instead, we propose a linear policy-based method, allowing us to tackle the scenario where the state is itself a function. In concrete terms, we introduce a family of policies, Functional Linear Policies (FLPs), which take continuous actions depending on functional states. These policies are direct adaptations of functional linear models (Cardot et al., 1999). Additionally, we prove that the main theoretical result from TRPO still holds for FLPs, and we adapt TRPO to propose practical algorithms to improve these policies. Prior work proposed the TRPO update in a classical RL context, with finite action and state spaces (Schulman et al., 2015a). Our main theoretical result follows the same proof of Schulman et al. (2015a, Theorem 1) and the proposed practical algorithms deal with issues arising from the highly dimensional setting by relying on classical FDA techniques: finite basis projection, FPCA (Wang et al., 2016) and a resolvent approach (Kreyszig, 2007). Additionally, based on these FDA techniques, we propose a

Proximal Policy Optimization (PPO) update. At last, we investigate the integration of classical functional basis reduction techniques (FDA) with modern policy optimization algorithms (TRPO/PPO) to achieve stable control in continuous linear PDE environments. These policies are interpretable, lightweight alternatives to policies parametrized by neural-networks, making them particularly suited to resource-constrained or embedded settings when the state is given by a function.

We would like to stress that, similar to previous work (Schweitzer & Seidmann, 1985; Bertsekas & Tsitsiklis, 1996; Konidaris et al., 2011; Brellmann et al., 2023; Li & Pathak, 2021) we also use a finite basis expansion, but our use is fundamentally different from the aforementioned works, as we focus on the *policy* side. In the prior literature, the state is a finite-dimensional vector $s \in \mathbb{R}^d$, and the basis functions $\phi_k : \mathbb{R}^d \to \mathbb{R}$ are nonlinear features *of* the state, used to approximate the value function: $V(s) \approx \sum_k \omega_k \phi_k(s)$. In our framework, the state itself is a function $s(\cdot) \in L^2([0,1])$, and writing $s(\cdot) = \sum_k a_k \phi_k(\cdot)$ expresses the state as an element of a Hilbert space, the coefficients $(a_k)_k$ are its coordinates. We then define the *policy* directly over these coordinates. Note that in our experiment our critic does take these coordinates as inputs to a neural network, similarly to prior work (Konidaris et al., 2011); however, this is not our contribution. Our contribution is the functional parameterization of the policy itself, which is what enables the Hilbert-space geometry underlying Proposition 1.

# 1 Background

## 1.1 RL in Borel Spaces

Let us consider an MDP $(\mathcal{S}, \mathcal{A}, q, r)$ where $\mathcal{S}$ and $\mathcal{A}$ are the state and action spaces, which we suppose are Borel spaces, $q$ is the transition law (probability measure on $\mathcal{S}$, given a state-action pair), and $r$ is the reward function. We note $\Pi$ the set of policies, which is defined as the set of probability distributions on $\mathcal{A}$, conditionally on a given state of $\mathcal{S}$. Hernandez-Lerma (2001) studied the construction of MDPs in this scenario.

Let us note $[\![1, n]\!]$ the set of integer numbers between 1 and $n$. Additionally, we note $\mathcal{B}(\mathcal{S})$ and $\mathcal{B}(\mathcal{A})$ the Borel sets of the state and action spaces, respectively.

Let $H_t$ be the set of rollouts up to time $t \in \mathbb{N}$ and $\Omega$ the set of (countably) infinitely long rollouts: $H_t = \{(s_0, a_0, s_1, a_1, \ldots, s_{t-1}, a_{t-1}, s_t) \mid s_k \in \mathcal{S}, a_k \in \mathcal{A}, k \in [\![1, t]\!]\}$ and $\Omega = \{(s_0, a_0, s_1, a_1, \ldots) \mid s_k \in \mathcal{S}, a_k \in \mathcal{A}, k \in \mathbb{N}\}$.

**Theorem 1** (Chapter 1, Section 2.5 Hernandez-Lerma (2001)). *Let $(\mathcal{S}, \mathcal{A}, q, r)$ be a MDP, let $(\Omega, \mathcal{F})$ be the measurable space where $\mathcal{F}$ is the product $\sigma$-algebra on $\Omega$. Then for every policy $\pi \in \Pi$ and for every initial state distribution $\rho$, there exists a unique probability measure $\mathbb{P}_\rho^\pi$ on $(\Omega, \mathcal{F})$ satisfying:*

*1. $\mathbb{P}_\rho^\pi(x_0 \in A) = \rho(A)$,*

*2. $\mathbb{P}_\rho^\pi(a_t \in B | h_t) = \pi(B | s_t)$,*

*3. $\mathbb{P}_\rho^\pi(s_{t+1} \in A | h_t, a_t) = q(A | s_t, a_t)$,*

*for all $A \in \mathcal{B}(\mathcal{S})$, $B \in \mathcal{B}(\mathcal{A})$, $h_t \in H_t$, $t \in \mathbb{N}$.*

In the following, for any $\mathcal{F}$-measurable function $h \colon \Omega \to \mathbb{R}$, we note $\mathbb{E}_\rho^\pi(h)$, the expectation of $h$, which is the Lebesgue integral, with respect to the probability measure $\mathbb{P}_\rho^\pi$, that is:

$$\mathbb{E}_\rho^\pi(h) = \int_\Omega h(\omega) \mathbb{P}_\rho^\pi(d\omega).$$

Let $\gamma \in ]0, 1[$ be a discount factor. The function $G = \sum_{t=0}^\infty \gamma^t r(s_t, a_t)$ is measurable and finite as long as the function $r$ is bounded. In the rest of this document, we suppose that the function $r$ is bounded on $\mathcal{S} \times \mathcal{A}$, thus expectation and sum may be interchanged. We can now consider the value, state-value, and advantage functions, defined respectively as $V^\pi(s) = \mathbb{E}_\rho^\pi(G | s_0 = s)$, $Q^\pi(s, a) = \mathbb{E}_\rho^\pi(G | s_0 = s, a_0 = a)$ and $A^\pi(s, a) = Q^\pi(s, a) - V^\pi(s)$.

## 1.2 Linear Functional Policies

Separable Hilbert spaces are Borel spaces and thus can be considered under the presented framework. A notable example of such a space is the space of square-integrable real functions, noted $L^2([0,1])$. The space $L^2([0,1])$ is a Hilbert space, endowed with the scalar product:

$$\langle f, g \rangle_{L^2} = \int_{[0,1]} f(s)g(s)ds \; ; \quad f, \; g \in L^2([0,1]).$$

In a supervised context, this enables the creation of a diverse range of function-to-scalar regression models, including functional linear models (Cardot et al., 1999), generalized functional linear models (Müller & Stadtmüller, 2005), and functional generalized additive models (McLean et al., 2014).

The same construction can be used to define policies over functional state spaces. Indeed, consider the parameters $\beta \in L^2([0,1])$, $c \in \mathbb{R}$, $\sigma \in \mathbb{R}^+$, then a Functional Linear Policy (FLP) is the conditional probability distribution with the following density:

$$\pi_\theta(a|s) = (2\pi\sigma^2)^{-1/2} \exp\left(-(c + \langle \beta, s \rangle_{L^2} - a)^2/2\sigma^2\right),$$

with $a \in \mathbb{R}$ and $s \in L^2([0,1])$.

Consider any initial state distribution $\rho(\cdot)$ and define the objective function: $J(\theta) = \mathbb{E}_\rho^{\pi_\theta}(G)$. The goal of policy optimization methods is to find the parameter that maximizes the objective function $J(\cdot)$. The same problem arises for FLPs, with $\theta = (\beta, c, \sigma) \in \Theta = L^2([0,1]) \times \mathbb{R} \times \mathbb{R}^+$. Note that FLPs are differentiable with respect to $\theta$: in the usual sense, for the real parameters $c$ and $\sigma$, and as a Fréchet derivative for the functional parameter $\beta$ (Hsing & Eubank, 2015, Section 3.6). Thus, it is possible to adapt policy gradient methods, such as REINFORCE (Sutton et al., 1999), NPG (Kakade, 2001), TRPO (Schulman et al., 2015a) or PPO (Schulman et al., 2017) in this framework.

## 1.3 TRPO

Let us consider the TRPO surrogate function, and the state average Kullback-Leibler divergence, defined respectively:

$$L_{\theta_{\text{old}}}(\theta) = \mathbb{E}_{\substack{s \sim \rho^{\theta_{\text{old}}} \\ a \sim \pi_{\theta_{\text{old}}}(\cdot|s)}} \left[ \frac{\pi_\theta(a|s)}{\pi_{\theta_{\text{old}}}(a|s)} Q^{\pi_{\theta_{\text{old}}}}(s,a) \right]$$

$$\bar{D}_{\text{KL}}(\theta_{\text{old}}||\theta) = \mathbb{E}_{s \sim \rho^{\theta_{\text{old}}}} \left(\text{KL}(\pi_{\theta_{\text{old}}}(\cdot|s), \pi_\theta(\cdot|s))\right).$$

TRPO is a policy gradient method designed for direct policy optimization. It achieves this by maximizing a surrogate objective $L_{\theta_{\text{old}}}(\cdot)$, which provides a first-order approximation of the true objective $J(\cdot)$ near $\theta_{\text{old}}$. To ensure stable updates, TRPO constrains the state-average Kullback-Leibler (KL) divergence between successive policies, $\bar{D}_{\text{KL}}(\theta_{\text{old}}||\cdot)$. The method is theoretically grounded in a performance improvement bound (Schulman et al., 2015a, Theorem 1). The TRPO update aims to solve the optimization problem (P):

$$\max_\theta L_{\theta_{\text{old}}}(\theta), \text{ such that } \bar{D}_{\text{KL}}^{\rho_{\theta_{\text{old}}}}(\theta_{\text{old}}, \theta) \leq C. \tag{P}$$

Schulman et al. propose to compute the direction of the update by solving a sample-based estimation of the objective and constraints, using a first-order approximation for the objective and a second-order approximation for the constraint. Once the step direction is computed, a line-search is performed to ensure that the surrogate effectively improves while respecting the KL constraint. Concretely, let us note $\delta$ the step given: $\theta = \theta_{\text{old}} + \delta$. For the "single path" procedure, $\delta$ is computed by solving the problem:

$$\max_\delta \langle g, \delta \rangle, \text{ such that } \langle \delta, \Gamma\delta \rangle \leq C. \tag{P approx}$$

where $g$ is the average gradient of the surrogate, $g = \mathbb{E}_{s \sim \rho_{\theta_{old}}}(\nabla_\theta L_{\theta_{old}}(\theta_{old}))$ and $\Gamma$ is the average expected information matrix: $\Gamma = \mathbb{E}_{s \sim \rho_{\theta_{old}}}\left(\nabla_\theta^2 \log \pi_{\theta_{old}}(\cdot|s)\right)/2$. For any value of $C$, there exists $\lambda > 0$ such that

Problem (P approx) is equivalent to the unconstrained problem maximize$_\delta \langle g, \delta \rangle - \lambda/2 \langle \delta, \Gamma \delta \rangle$, whose solution $\delta^*$, verifies:

$$g = \lambda \Gamma \delta^*. \tag{1}$$

### 1.4 Proximal Policy Optimization

TRPO is a policy optimization method of second order, as it relies on a Kullback-Leibler constraint. Inspired by this method, Schulman et al. (2017), proposed Proximal Policy Optimization (PPO), a first-order policy optimization method, similar to the TRPO method. Instead of approaching a maximization problem under constraints, a clipped objective is proposed that enforces that the new policy is close to the current policy, in a certain sense. More specifically, let $\theta_{\text{old}}$ be the current parameter, and consider the observed probability ratio at instant $t$: $r_t(\theta) = \pi_\theta(a_t|s_t)/\pi_{\theta_{\text{old}}}(a_t|s_t)$, the authors propose the following objective:

$$\mathbb{E}\left[\min\left(r_t(\theta)\hat{A}_t, \text{clip}(r_t(\theta), 1 - \epsilon_{\text{clip}}, 1 + \epsilon_{\text{clip}})\right)\hat{A}_t\right],$$

where $\text{clip}(x, a, b) = \min(\max(x, a), b))$ denotes the clipping operator, $\hat{A}_t$ is an estimate of the advantage function at the moment $t$ and $\epsilon_{\text{clip}} > 0$ is a hyperparameter; a common value is $\epsilon_{\text{clip}} = 0.2$. The intuition, given by Schulman et al. (2017), is that for a given timestep $t$, if the new policy is close to the current one (e.g., $r_t(\theta) \in [1 - \epsilon_{\text{clip}}, 1 + \epsilon_{\text{clip}}]$), then the clipped objective is the same as the TRPO surrogate; otherwise, the second term $\text{clip}(r_t(\theta), 1 - \epsilon_{\text{clip}}, 1 + \epsilon_{\text{clip}})$ removes the incentive of taking a step outside or the interval $[1 - \epsilon_{\text{clip}}, 1 + \epsilon_{\text{clip}}]$, as the final objective is the minimum of those two.

## 2 TRPO and PPO for FLPs

The main objective of this paper is to adapt TRPO and PPO to improve FLPs. The following result is the main theoretical contribution of this contribution; it allows us to propose practical TRPO methods.

**Proposition 1.** *Let $\pi_\theta$, $\pi_{\tilde{\theta}}$ be two FLPs with respective parameters $\theta = (\beta, c, \sigma) \in \Theta$ and $\tilde{\theta} = (\tilde{\beta}, \tilde{c}, \tilde{\sigma}) \in \Theta$, and consider $M > 0$. If the state space is bounded: $\mathcal{S} = \{f \in L^2([0,1]) \mid \|f\|_{L^2} \leq M\}$. Then the function $s \mapsto D_{KL}(\pi_{\tilde{\theta}}(\cdot|s)||\pi_\theta(\cdot|s))$ is continuous, attains a maximum in $\mathcal{S}$ and the function $(s, a) \mapsto |A^{\pi_\theta}(s, a)|$ is bounded. Let $\alpha^2 = \max_{s \in \mathcal{S}} D_{KL}(\pi_\theta(\cdot|s)||\pi_{\tilde{\theta}}(\cdot|s))$ and $\epsilon = \sup_{s,a}|A^{\pi_\theta}(s, a)|$. The following bound holds:*

$$J(\tilde{\theta}) \geq L_\theta(\tilde{\theta}) - 4\epsilon\gamma\alpha^2/(1 - \gamma)^2. \tag{2}$$

*Proof.* We provide a detailed proof in the Annex A. □

As stated in Proposition 1, the improvement bound from (Schulman et al., 2015a, Theorem 1) still holds when the state space is the set of bounded functions $L^2([0,1])$ and the family of policies considered are FLPs. Using this result, just as done by Schulman et al. (2015a), we derive a practical algorithm by replacing the maximum on the KL constraint by the state-average KL, obtaining the same optimization problem stated in Problem (P), except the parameter space is the infinite-dimensional space $\Theta = L^2([0,1]) \times \mathbb{R} \times \mathbb{R}^+$. Using sample-based estimations, the direction is computed using a first order approximation of the surrogate and a second order approximation of the state-average KL constraint. Thus, we obtain optimization Problem (P approx), except that the gradient of the surrogate $g$ is not a vector but an element of $\Theta$ and $\Gamma$ is not a matrix but an operator. The solution to this problem, $\delta^* \in \Theta$, still verifies (1).

Yet, a difficulty arises for the implementation of a practical algorithm: Equation (1) is a linear inverse problem in an infinite-dimensional space, the operator has no global inverse, and its pseudo-inverse may not even be continuous.

Luckily, this problem is common in the FDA literature, and there exist common techniques to work around this difficulty. Concretely, we implement and compare four methods: using a finite basis expansion (Section 2.1), FPCA (Section 2.2) or using a resolvent estimator (Section 2.3).

### 2.1 Naïve approach

A common technique in FDA (Wang et al., 2016) is to consider a finite expansion of the functional terms. Let $(\phi_i(\cdot))_{i\in\mathbb{N}}$ and $(\psi_j)_{j\in\mathbb{N}}$ be functional bases of $L^2([0,1])$. Consider two functions $f, g \in L^2([0,1])$, and suppose they can be written using $N_\psi$ and $N_\phi$ terms from the functional bases, respectively. Then they are of the following form: $f(\cdot) = \sum_{i=1}^{N_\Psi} a_i \cdot \psi_i(\cdot) = \boldsymbol{a}^T \cdot \boldsymbol{\Psi}(\cdot)$ and $g(\cdot) = \sum_{j=1}^{N_\Phi} b_j \cdot \phi_j(\cdot) = \boldsymbol{b}^T \boldsymbol{\Phi}(\cdot)$, where $\boldsymbol{a} = (a_1, \ldots, a_{N_\Psi})^T$ and $\boldsymbol{b} = (b_1, \ldots, b_{N_\Phi})$ are vectors of coefficients, and $\boldsymbol{\Psi}(\cdot) = (\psi_1, \ldots, \psi_{N_\Psi})$ and $\boldsymbol{\Phi}(\cdot) = (\phi_1, \ldots, \phi_{N_\Psi})$ are vectors of functions. Then the functional scalar product between $f$ and $g$ can be written as a matrix product: $\langle f, g \rangle_{L^2} = \boldsymbol{a}^T \boldsymbol{R} \boldsymbol{b}$, where $\boldsymbol{R}$ is the $N_\phi \times N_\psi$ matrix with the inner product of the elements from the two functional bases, $\boldsymbol{R} = (\langle \phi_i, \psi_j \rangle)_{i\in[\![1,N_\Phi]\!], j\in[\![1,N_\Phi]\!]}$.

This can be used to turn TRPO for FLPs into a finite-dimensional problem: instead of computing $\delta(\cdot) \in L^2([0,1])$, if the step can be written in a finite functional basis: $\delta(\cdot) = \boldsymbol{b}^T \boldsymbol{\Phi}(\cdot)$, then it is enough to compute its functional coefficients, $\boldsymbol{b} \in \mathbb{R}^{N_\Phi}$.

### 2.2 FPCA approach

FPCA provides a functional basis of orthonormal eigenfunctions. Let, $u, v \in [0,1]$, we consider the functional states $s(\cdot)$, which are random curves following a distribution $\rho$. Let us note $\mu_\rho(u) = \mathbb{E}_{s\sim\rho}(s(u))$ the functional mean and $\Sigma_\rho(u,v) = \mathbb{E}_{s\sim\rho}((s(u) - \mu(u))(s(v) - \mu(v)))$ the functional covariance.

The covariance operator $\Sigma_\rho(f(\cdot)) = \int \Sigma(\cdot,u)f(u)du$ is a positive, symmetric, compact operator. By the Karhun-Loeve theorem, there exists an orthonormal functional basis $(\xi_k(\cdot))_{k\in\mathbb{N}}$, with corresponding eigenvalues, $(\lambda_k)_{k\in\mathbb{N}}$ such that: $\Sigma(u,v) = \sum_{k\in\mathbb{N}} \lambda_k \xi_k(u)\xi_k(v)$ and $\langle \xi_j, \xi_k \rangle_{L^2} = \mathbb{1}_{j=k}$, for all $j, k \in \mathbb{N}$. FPCA is particularly relevant for TRPO: if the functional mean of $s$, with respect to $\rho$ is null, the quadratic constraint of Problem (P approx) becomes diagonal. Indeed, let $\theta = (\beta(\cdot), c, \sigma) \in \Theta$ and $\delta = (\delta_\beta(\cdot), \delta_c, \delta_\sigma) \in \Theta$, consider a state $s \in \mathcal{S}$, then the explicit form for state-average KL divergence is:

$$\bar{D}_{\mathrm{KL}}(\theta || \theta + \delta) = \log\left(\tfrac{\sigma+\delta_\sigma}{\sigma}\right) + \tfrac{\sigma^2 + \langle \delta_\beta, \Gamma_\Sigma(\delta_\beta)\rangle^2_{L^2} + \delta_\sigma^2}{2(\sigma+\delta_\sigma)^2} - \tfrac{1}{2}. \tag{3}$$

Projecting $\delta_\beta(\cdot)$ upon the first $N_\xi \in \mathbb{N}$ components allows us to efficiently use a low rank approximation of the Kullback-Leibler divergence. Moreover, in the eigenbasis $(\xi_k)_{k=1}^{N_\xi}$, the second order approximation of the constraint is a diagonal quadratic form, with respect to the coefficients of $\delta_\beta(\cdot)$. Indeed, let us suppose $\delta_\beta(\cdot) = \sum_k^{N_\xi} b_k \xi_k(\cdot)$, then Equation 3 becomes:

$$\bar{D}_{\mathrm{KL}}(\theta || \theta + \delta) = \log\left(\tfrac{\sigma+\delta_\sigma}{\sigma}\right) + \tfrac{\sigma^2 + \sum_k^{N_\xi} \lambda_k b_k^2 + \delta_\sigma^2}{2(\sigma+\delta_\sigma)^2} - \tfrac{1}{2}, \tag{4}$$

which is a diagonal quadratic form in the functional coefficients $(b_k)_{k=1}^{N_\xi}$.

It is important to note that the eigenfunctions depend on the state distribution, $\rho$ and thus the FPCA decomposition should be computed after every episode. In practice, to select the number of eigenfunctions, we keep only the ones with sufficiently large eigenvalues.

### 2.3 Resolvent approach

An alternative solution to equation 1 can be obtained through resolvents. The solution by projection is equivalent to approximating $\Gamma^{-1}$ using a linear operator with additional regularity, $\Gamma^\dagger = \sum_{k=1}^{N_\xi} b(\lambda_k)(\xi_k \otimes \xi_k)$, where $N_\xi$ is an increasing sequence of integers tending to infinity, and $b$ is a smooth function converging pointwise to $x \mapsto 1/x$. Indeed, $\Gamma^\dagger \to \Gamma^{-1}$ as $N_\xi \to \infty$. Choosing $b(x) = 1/x$ for a finite $N_\xi$ results in setting $\Gamma^\dagger$ to be a spectral cutoff approximation of $\Gamma^{-1}$. However, this choice is not unique. Consider the following family of functions $b_{n,p} : \mathbb{R}_+ \to \mathbb{R}_+$, parameterized by $p \in \mathbb{N}$, such that

$$b_{n,p}(x) = x^p / (x + \alpha_n)^{p+1},$$

where $\alpha_n$ is a strictly positive sequence that tends to 0 as $n \to \infty$.

Based on this, we can define the following class of solutions

$$\delta_p^* = b_p(\Gamma)g, \tag{5}$$

where $b_p(\Gamma) = (\Gamma + \alpha I)^{-(p+1)}\Gamma^p$, with $p \geq 0$, $\alpha > 0$. This resolvent class corresponds to regularized approximations of $\Gamma^{-1}$ to address the inversion problem (Martini et al., 2022).

### 2.4 PPO for FLPs

We readily adapt the PPO update using the functional methods of Sections 2.1 and 2.2. We compute the observed probability ratio in terms of the functional scalar product, which can be computed using a finite basis expansion—for a given and known functional basis as explained in Section 2.1, or as a basis of eigenfunctions, as presented in Section 2.2. We cannot adapt the resolvent approach for PPO, as the resolvent approach considers a second method, but PPO is a first-order method.

## 3 Numerical Experiments

In this section we aim to answer the following questions:

1. Do our proposed methods work in practice?

2. How do they compare among themselves?

3. How do they compare against taking the raw observations and using standard DRL methods?

We do the best we can to answer these carefully by following guidelines presented by Patterson et al. (2024). We compare RL methods rigorously, using powerful statistical tests.

In theory, our methods should work well when the states are measurements from an underlying inherently functional state, such as the environments developed and implemented in `controlgym` python package (Zhang et al., 2024). Furthermore, in some of these environments, the optimal policy can be computed if the environment is completely known and perfectly observed, which provides a sort of ideal baseline for our RL methods.

In addition to this section's numerical experiments, in the Appendix E, we investigate the impact of an irregular grid on the performance of FLPs.

### 3.1 PDE control environments

Let us first present the true continuous controlled PDE problem, which we then formulate as an RL problem with a functional state space.

Just as presented by Zhang et al. (2024), we consider a "one-dimensional PDE control environment with periodic boundary conditions and spatially distributed control inputs". We consider a spatial domain $[0, 1]$, let $T > 0$, and a continuous field $u\colon [0, 1] \times [0, T] \to \mathbb{R}$, where $u(\cdot, t)$ is the state function at an instant $t$. This continuous field evolves according to the a controlled PDE:

$$\frac{\partial u}{\partial t} - \mathcal{L}\left(\frac{\partial u}{\partial x}, \frac{\partial^2 u}{\partial^2 x}, \dots\right) = a(x, t), \tag{6}$$

with $x \in [0, 1]$, $t \in [0, T]$ and $\mathcal{L}$ is a linear differential operator and $a(x, t)$ is the distributed control force applied at location $x$ at instant $t$.

Unlike what was proposed by Zhang et al. (2024), in this paper, we consider only a single scalar input, applied over the subdomain $[0.3, 0.7]$, that is, we suppose it is given by $a(x, t) = \mathbb{1}_{[0.3, 0.7]}(x)a(t)$.

Let us consider a given ideal state function $u_{\text{ideal}}(\cdot) \in L^2([0,1])$, we wish to find a distributed control, $a(\cdot)$, such that the following quantity is minimized:

$$\int_{[0,T]} \int_{[0,1]} (u(x,t) - u_{\text{ideal}}(x))^2 dx dt + \int_{[0,T]} a(t)^2 dt. \tag{7}$$

This problem can be formulated as an RL problem by discretizing the equation in time, with functional states and actions sampled from an FLP $\pi_\theta$. For each time step $t_i$, the state observation is given by $s_i(\cdot) = u(\cdot, t_i) \in L^2([0,T])$, the action is the input applied at the instant $t_i$, which in turn is sampled from an FLP $a_i \sim \pi_\theta(\cdot|s_i)$ and the reward is defined by $r_i = -\int_{[0,1]} (s_i(x) - u_{\text{ideal}}(x))^2 dx - \lambda a_i^2$.

In practice, this controlled PDE is also discretized in space; we consider a fine grid with $N_x \in \mathbb{N}$ points $\{x_k\}_{k=1}^{N_x}$, at the timestep $t_i$, we compute the states only on points $s_i(x_k) \in \mathbb{R}$. In the package `controlgym`, these are computed using efficient numerical methods. At last, to mimic real-world scenarios, we only observe $0 < N_{\text{obs}} \leq N_x$ noisy versions of some of these points. Concretely, let $\boldsymbol{s}_i = (s_i(x_1), \ldots, s_i(x_{N_x}))$ be the vector of states at the discretization points; we observe a vector of noisy measurements $\boldsymbol{s}_i^{\text{obs}} \in \mathbb{R}^{N_{\text{obs}}}$

$$\boldsymbol{s}_i^{\text{obs}} = \boldsymbol{C}\boldsymbol{s}_i + \boldsymbol{e}_i \; ; \; \boldsymbol{e}_i \sim \mathcal{N}(\boldsymbol{0}_{N_{\text{obs}}}, \boldsymbol{\Sigma}), \tag{8}$$

where $\boldsymbol{C} \in \mathbb{R}^{N_{\text{obs}} \times N_x}$ is a matrix with a single 1 per row and zeros elsewhere, and the covariance matrix $\boldsymbol{\Sigma}$ is user-specified.

Although the state function $s_i(\cdot)$ is not truly observed, there exists an extensive literature on how to estimate it using the noisy discrete observations $\boldsymbol{s}_i^{\text{obs}}$, the main idea being choosing the functional coefficients in such a way that the mean-square error is minimized while penalizing the roughness of the obtained functional object. The reader may consult Ramsay & Silverman (2005) or Kreyszig (2007) for more details.

## 3.2 Method

We evaluate and compare our methods in all the linear PDE environments from `controlgym`, as these allow comparing performances against the ideal LQR controllers. Namely, we carry out our experiments in the three following environments: "wave", "convection_diffusion_reaction" and "schrodinger". We consider episodes with 100 temporal steps, with a temporal discretization step of 0.1, the spatial domain is discretized in 16 points, and observed only on 12 points. The rest of the parameters are the default-ones proposed in `controlgym`, and are specific for each PDE environment. As an intuitive baseline, we also compare our methods to the zero-controller, whose action is always 0.

We use a discount rate $\gamma = 0.99$ for all experiments, and, as recommended by Huang et al. (2022a), we use generalized advantage estimation (Schulman et al., 2015b), with weight $\lambda_{GAE} = 0.95$ in all experiments. This allows us to estimate the advantage function the same way for all methods.

For our functional methods, we use a Fourier basis, with 10 elements, to parametrize both the $\beta(\cdot)$ and the state functions. We use the implementation from the Python package `scikit-fda` (Ramos-Carreño et al., 2024). The $\beta(\cdot)$ parameter is initialized as the null-function and the constant $c$ is initialized as 0; thus, during the first episode, actions are taken independently of the observed state. Additionally, we implement a critic network given by a fully connected feedforward neural network taking as input the functional coefficients of the state function and having two hidden layers of size 64. Each hidden layer of the critic is followed by a hyperbolic tangent activation function.

To compare against our methods, we consider the PPO controller implemented in the `controlgym` package. This controller uses a classical actor-critic structure, using an actor and a critic with independent networks, both parametrized by a multilayer perceptron, with ReLu activations and two hidden layers of size 64. All linear layers are initialized orthogonally as suggested by Saxe et al. (2014) with $\sqrt{2}$ gain and biases initialized as zero. This network takes as input the discrete values $\boldsymbol{s}_i^{\text{obs}} \in \mathbb{R}^{12}$.

Both the FLPs and the neural-network policy have a standard deviation parameterized as a state-free learnable parameter, which is initialized as $\sqrt{0.05}$, to encourage small early action outputs.

We adapt the PPO implementation from the `cleanRL` package (Huang et al., 2022b), which follows guidelines proposed by Huang et al. (2022a). We rely on automatic differentiation, implemented in `PyTorch` (Paszke et al., 2019). For each method, we run 20 episodes in parallel, and using these episodes, perform one PPO update. We repeat this until we obtain 4000 episodes; this defines one learning trajectory.

To provide a fair comparison between methods, as suggested by Patterson et al. (2024), we carefully tune each method separately in each environment. For a given environment, we test 20 combinations of hyperparameters, running three seeds per set of hyperparameters, doing an efficient search using the `optuna` package (Akiba et al., 2019). In addition to the functional methods hyperparameters, we tune for the learning rate in PPO and both the critic learning rate and the maximum KL in TRPO. We provide a summary of the range of search of hyperparameters as well as the selected hyperparameter for each environment and method, as well as other parameters (number of gradient descents, batch size, etc.), in Annex B.

At last, we evaluate the performance of each tuned method, in each environment and method, by running 50 learning trajectories with different random seeds. As suggested by Patterson et al. (2024), we use the same random seeds across methods, allowing us to control the initial state across methods. Doing this, we obtained paired measurements, which we used to compare methods using paired statistical tests, which are more powerful than unpaired tests.

All experiments were conducted on a machine with an Intel(R) Xeon(R) Silver 4114 CPU and an NVIDIA RTX A6000 GPU. The complete set of experiments, including tuning and comparisons across all methods and environments, required approximately 330 hours of compute. We refer to Annex C for a detailed outline of the algorithm and Annex F for a discussion on the computational complexity.

### 3.3 Results

In Figure 1, we present the evolution of the median episodic reward, calculated using the 50 random seeds. For comparison, we show the performances of the LQR and zero controllers. For visualization purposes, values smaller than baseline minus ten are truncated and displayed at the threshold. Additionally, we show the two-sided 0.95-confidence interval for the median, calculated using bootstrap. We observe that updating FLPs using any proposed method, either using PPO or TRPO, improves the performance of the functional policy. Indeed, at the end of training, the median episodic reward of FLPs is significantly better than those of the zero controller. Additionally, we note that functional methods perform similarly for a given algorithm, except when using PPO in the "convection" environment, where FPCA seems to outperform the naive method at the end of training.

In contrast, the performance of the Neural Network (NN) policy is not significantly better than the performance of the zero controller in any environment. Interestingly, for environments "convection" and "wave", using TRPO to improve the NN policy *sometimes* yields better performances than the zero controller, as the point estimate is above that baseline, but sometimes it is way lower than the zero controller, as the lower confidence interval is lower than the zero controller. With PPO in environments "convection" and "schrodinger", the NN policy improves stably but never above the zero controller, and in environment "convection" performances increase until they attain those of the zero controller. To sum up, updating the NN policy with PPO yields ineffective policies, and updating it with TRPO yields unstable performances. In Figure 2 we present the evolution of the median difference of episodic rewards of FLPs and the NN policy. Additionally, we show the one-sided 0.95-confidence interval for the median, calculated using bootstrap. For visualization purposes, we clip the median difference between policies and its lower bound at 50. We observe that using PPO with any functional method yields performances that are significantly better than using the NN policy, all through training. Similarly, when using TRPO, in the environments "schrodinger" and "wave", we observe that all the functional methods outperform the NN policy, and in the environment "convection", the resolvent approach does not seem significantly better than the NN policy.

At last, in Figure 3, we compare the performances of methods at the end of training. For each method and random seed, we compute the average reward over the last 20 episodes, then clip the obtained average at -500. This yields one scalar measurement of performance. We show the probability density of this quantity for each method and algorithm in each environment, estimated with the scalar measurements obtained from the 50 random seeds. Note that the x-axis is in the $\log 10$ scale. For reference, we add vertical lines with the

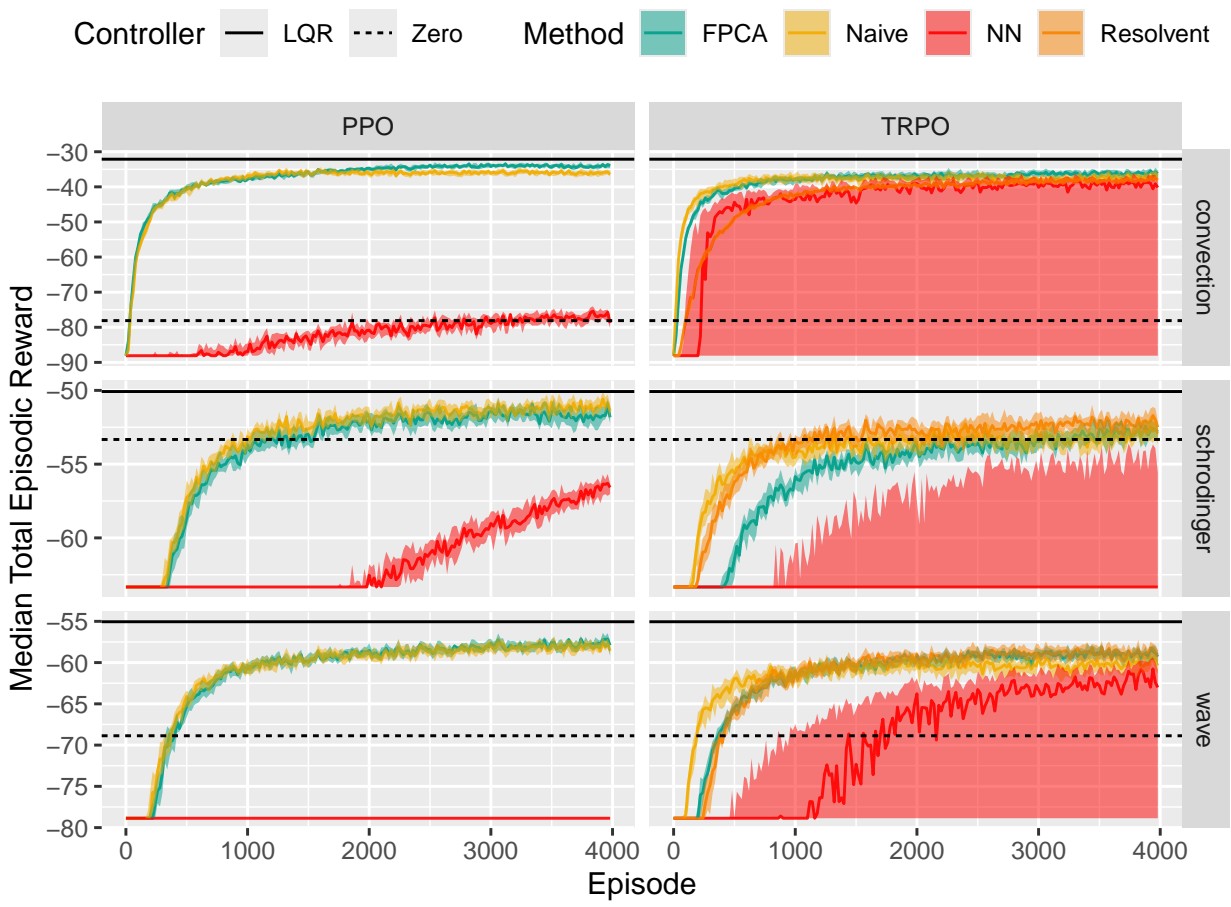

Figure 1: Median Episodic Reward vs. Episode for Different Algorithms and Methods.

performances of the LQR and zero controllers. As we observed in Figures 1 and 2, we can see that trained FLPs performances are better than the zero controller and usually are close to the ones of the ideal LQR controller. For NN policies, we observe that most of the density of the reward after training lies below the performance of the zero controller when using PPO and the density is bimodal when using TRPO—some trained NN policies will perform better than the zero-controller but some have bad performances.

To further evaluate the final performance of models, in each environment and for each algorithm, we compare them using paired non-parametric tests. We first evaluate if we observe a significant difference among methods, using the Friedman test (Friedman, 1937), and if there is one, we perform post-hoc comparisons using Wilcoxon's test (Mann & Whitney, 1947), correcting the p-values for the multiple testing using the Benjamini-Hochberg procedure (Benjamini & Hochberg, 1995). We observe that functional methods significantly outperform the NN policy in all environments when using PPO. Likewise, the naive and FPCA functional methods significantly outperform the NN policy in all environments, and so does the resolvent method in all environments except in the "convection" environment; in this last environment, the resolvent method is outperformed by all the other methods. There is no significant difference between the FPCA and naive functional methods. We provide all the pairwise comparisons in Annex C.

To answer our initial questions:

1. Our proposed methods do seem to work for PDE control; as policies are updated, we do observe an increase in the episodic rewards. After training, all our methods perform significantly better than the zero-controller.

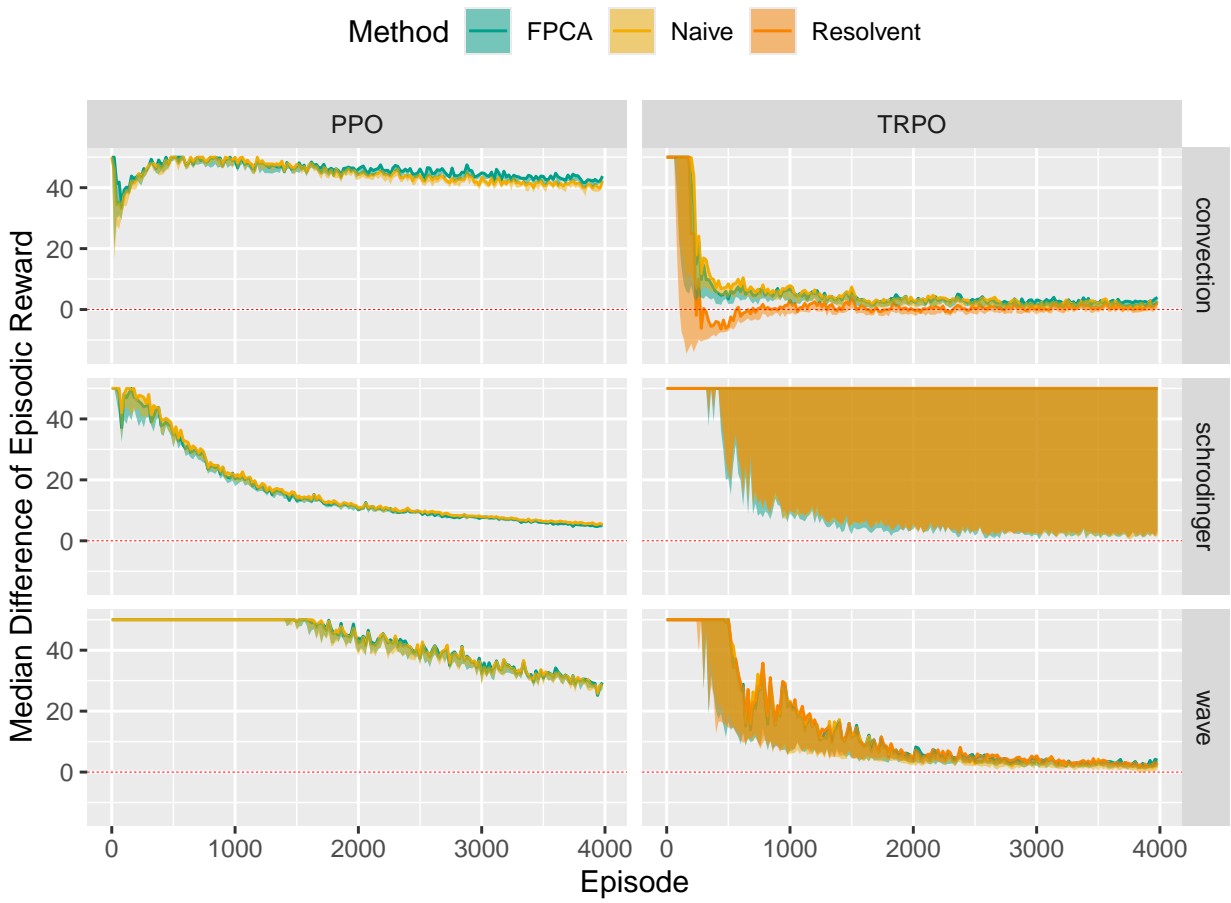

Figure 2: Difference Between the Median Episodic Reward of FLPs and Neural-Network Policy vs. Episode.

2. All of our functional methods perform similarly: the naive method is as effective as more intricate methods (FPCA or resolvent), suggesting that the tested `controlgym` environments likely possess a low intrinsic dimensionality. And using the PPO algorithm yields similar or better performances than TRPO, which is coherent with the literature (Schulman et al., 2017).

3. We observed that using FLPs instead of a MLP parametrized policy yielded significantly better performances in these environments.

## 4  Limitations

While our theoretical framework and algorithmic developments are general, the numerical experiments in this paper were carried out only on three environments. Indeed, we focused on canonical *linear* PDE control environments as a first testbed, which already present significant challenges while offering explicit, interpretable baselines. In this practical setting, we proposed a class of policies taking actions depending on a functional state and proved that TRPO succeeds in improving such policies, whereas standard NN-parameterized policies did not yield competitive performance in this setting. Extending to broader application domains is an exciting future direction. The method extends to mildly nonlinear dynamics: although the policy remains a functional linear model (acting as a first-order functional approximation), it can still be applied to nonlinear systems in a manner analogous to how standard linear policies are used in classical RL for complex, nonlinear continuous control tasks. While current state-of-the-art physics-informed machine learning frameworks rely

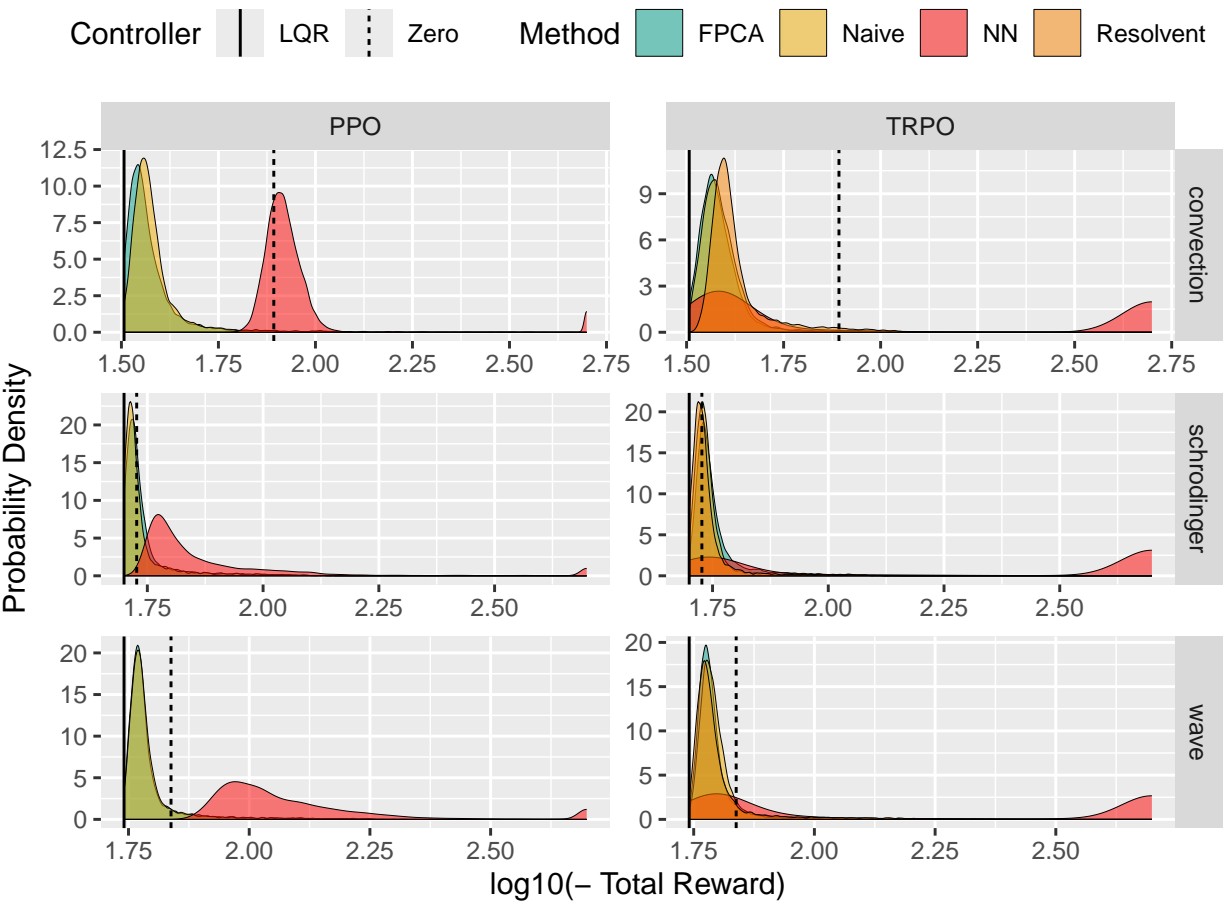

Figure 3: Probability Densities of Final Episodic Reward for Different Algorithms and Methods.

on architectures like Neural Operators for PDE control, first-order functional linear approximations remain highly valuable due to their inherent interpretability and significantly lower computational overhead.

In our experiments we used an MLP-parameterized policy, as these are the classic choices in the RL literature. While such architecture does not encode spatial structure, it provides a standard baseline for comparison. Fundamentally, an MLP lacks a spatial inductive bias; it treats discrete spatial observations as independent, isolated inputs without an inherent awareness of the underlying grid's continuity. When physical phenomena shift between sparse sensors, the MLP perceives disjointed scalar fluctuations rather than a coherent moving wave. While alternative architectures like recurrent neural networks or explicit positional encodings could introduce spatial awareness, they often require higher sample complexity to resolve the underlying physical field. In contrast, our framework leverages Functional Data Analysis (FDA) to project these sparse measurements directly onto a continuous functional basis, effectively reconstructing the global physical state. This achieves structural parameter economy by scaling the policy strictly with the complexity of the underlying functional representation (see Appendix F), enhancing sample efficiency. Furthermore, standard MLPs optimize weights without respecting the continuous geometry of the state space, meaning small numerical adjustments can lead to erratic, non-smooth control actions that destabilize training. By enforcing a functional operator structure within our trust-region updates, our method ensures that the policy's physical behavior shifts smoothly across iterations, providing the numerical robustness required to successfully control the partial differential equation (PDE) dynamics. Exploring architectures better suited to functional states may further improve performance, but this lies outside the scope of our focus on FLPs.

In this paper, we considered linear functional methods; these may not be as expressive as deep RL methods; nevertheless, an immediate extension allowing for more expressive policies is using functional additive models (McLean et al., 2014). We considered the scenario of a univariate functional state; when the state is a multivariate functional space, using the signature representation (Kidger et al., 2019; Cugliari et al., 2025) may prove more effective than our functional approach.

## Conclusion and further work

This work introduces a family of policies capable of handling functional state spaces and extends TRPO to provide a practical update algorithm for such policies. These methods offer a promising approach to addressing high-dimensional state representations, particularly in spatial and temporal settings. While this study marks an initial step, it opens several avenues for further development and real-world applications.

Deep reinforcement learning has proven efficient in numerous complex tasks, but RL methods do not necessarily need to use deep learning, and for some applications, simpler models may be preferred. Indeed, at inference time, an FLP requires only a dot product between the basis coefficients of the state and the learned parameter $\beta(\cdot)$, which is substantially cheaper than a forward pass through a multi-layer perceptron. For instance, using simpler models may be particularly relevant for embedded or wearable devices, where compute, memory, and battery are limited. Anonymous et al. (Year)[1] illustrate this point with a concrete example: using a functional regression model provides competitive performance while consuming less energy; later, Anonymous et al. (Year)[1] demonstrated that using an RL approach seemed effective for the personalized control of a wearable device. This present work extends RL to account for functional states, filling a gap in the literature and enabling a wide range of applications, such as the personalized control of wearable devices using functional inputs.

A key advantage of using FDA representations is their ability to handle missing values and irregular sampling—common challenges in real-world problems. As we show in the experiments from Appendix E, FLPs do seem to generalize well in the presence of missing values or irregular grids. In contrast, traditional methods such as state stacking may struggle in this setting.

Further generalizations include exploring function-to-scalar policies for discrete or ordinal actions or even function-to-function policies, enabling functional actions, which could be used to approach control problems where the control itself is a function, from an RL standpoint. Notably, the framework of Hernandez-Lerma (2001) accommodates any state and action spaces that are Borel spaces, suggesting broader applicability.

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

## A    Appendix: proof of Proposition 1

To prove Proposition 1, we rely on Lemma 6.1 from Kakade & Langford (2002) and Lemma 3 of Schulman et al. (2015a), their original proof still holds in the context of this article. For completeness, we remind them in Lemma 1 and Lemma 2 respectively. The main idea is that the objective function $J(\cdot)$ can be approximated, locally in the current parameter $\tilde{\theta}$ by a surrogate $L_{\tilde{\theta}}(\cdot)$, which can then be improved; the main result of TRPO (Schulman et al., 2015a), bounds the difference between these two.

**Lemma 1.** *Given two policies parametrized by $\theta$ and $\tilde{\theta}$:*

$$J(\tilde{\theta}) = J(\theta) + \mathbb{E}_{\tilde{\theta}}\left(\sum_{t=0}^{\infty} \gamma^t A_\theta(s_t, a_t)\right).$$

**Definition 1.** *A couple $(\pi, \tilde{\pi})$ of policies are $\alpha$-coupled if their joint distribution verifies $\mathbb{P}_{a \sim \pi(\cdot|s), \tilde{a} \sim \tilde{\pi}(\cdot|s)}(a \neq \tilde{a}|s) \leq \alpha$, for all $s \in \mathcal{S}$*

**Lemma 2.** *Let $(\pi, \tilde{\pi})$ be $\alpha$-coupled, then:*

$$|\mathbb{E}_{s_t \sim \tilde{\pi}}(\bar{A}(s_t)) - \mathbb{E}_{s_t \sim \pi}(\bar{A}(s_t))| \leq 4\alpha(1 - (1-\alpha)^t) \sup_{s,a} |A_\pi(s,a)|. \tag{9}$$

*Proof of Proposition 1.* Let $\theta, \tilde{\theta} \in \Theta$ and $s \in \mathcal{S}$, using the explicit form of the Kullback-Leibler between two random normal distributions we obtain:

$$D_{\mathrm{KL}}(\pi_\theta(\cdot|s)||\pi_{\tilde{\theta}}(\cdot|s)) = \log\left(\frac{\tilde{\sigma}}{\sigma}\right) + \frac{\sigma^2}{2\tilde{\sigma}^2} + \frac{|\langle \beta - \tilde{\beta}, s \rangle_{L^2}|^2}{2\tilde{\sigma}^2} - \frac{1}{2}. \tag{10}$$

Thus, $s \mapsto D_{\mathrm{KL}}(\pi_\theta(\cdot|s)||\pi_{\tilde{\theta}}(\cdot|s))$ is a continuous function, and because $\mathcal{S}$ is bounded, the function $s \mapsto D_{\mathrm{KL}}(\pi_\theta(\cdot|s)||\pi_{\tilde{\theta}}(\cdot|s))$ attains a maximum in $\mathcal{S}$. The function $(s,a) \mapsto |A^{\pi_\theta}(s,a)|$ is bounded because the reward function is bounded. Thus, $\alpha^2 = \max\limits_{s \in \mathcal{S}} D_{\mathrm{KL}}(\pi_{\tilde{\theta}}(\cdot|s)||\pi_\theta(\cdot|s))$ and $\epsilon = \sup\limits_{s,a}|A^{\pi_\theta}(s,a)|$ are finite and well-defined.

The rest of the proof follows the same lines as the one provided in Annex of Schulman et al. (2015a).

Consider $\bar{A}(s) = \mathbb{E}_{a \sim \pi_{\tilde{\theta}}}(A_{\pi_\theta}(s,a))$, by Lemma 1, we can write:

$$J(\tilde{\theta}) = J(\theta) + \mathbb{E}_{\tilde{\theta}}\left(\sum_{t=0}^{\infty}\gamma^t \bar{A}(s_t)\right) \quad ; \quad L_\theta(\tilde{\theta}) = J(\theta) + \mathbb{E}_\theta\left(\sum_{t=0}^{\infty}\gamma^t \bar{A}(s_t)\right).$$

Let $D_{TV}$ be the total-variation between probability distributions. By Pinsker's inequality, for any state $s \in \mathcal{S}$, we have $D_{\mathrm{TV}}(\pi_\theta(\cdot|s), \pi_{\tilde{\theta}}(\cdot|s))^2 \leq D_{\mathrm{KL}}(\pi_\theta(\cdot|s)||\pi_{\tilde{\theta}}(\cdot|s))$, thus $D_{\mathrm{TV}}(\pi_\theta(\cdot|s), \pi_\theta(\cdot|s)) \leq \alpha$. By (Levin & Peres, 2017, Proposition 4.7), there exists a $\alpha$-coupling of the policies $(\pi_\theta, \pi_{\tilde{\theta}})$. By Lemma 2, we obtain:

$$|J(\tilde{\theta}) - L_\theta(\tilde{\theta})| = \sum_{t=0}^{\infty}\gamma^t|\mathbb{E}_{s_t \sim \tilde{\pi}}(\bar{A}(s_t)) - \mathbb{E}_{s_t \sim \pi}(\bar{A}(s_t))| \tag{11}$$

$$\leq \sum_{t=0}^{\infty}\gamma^t 4\alpha(1 - (1-\alpha)^t)\epsilon \tag{12}$$

$$= \frac{4\epsilon\alpha^2\gamma\epsilon}{(1-\gamma)(1-\gamma(1-\alpha))} \leq \frac{4\gamma\epsilon\alpha^2}{(1-\gamma)^2} \tag{13}$$

$\square$

# B  Appendix: details of numerical experiments of Section 3

Note that not all of our methods have the same number of hyperparameters; then, to provide a fair comparison, as suggested by Patterson et al. (2024), each method is tuned using only 20 trials in total for the hyperparameter search.

Independently of the method, for the critic training in TRPO and for the PPO update, we used 10 gradient steps, splitting the batch in 4 mini batches.

We show the critic learning rates, $\gamma_{\mathrm{critic}}$, and maximum Kullback Leibler divergences, $C_{\mathrm{KL}}$, for the TRPO update in Table 1. The range of search for $\gamma_{\mathrm{critic}}$ is [5.0e-05, 5.0e-03] and for $C_{\mathrm{KL}}$ is [1.0e-04, 1.0e-01]. As suggested by Patterson et al. (2024), the search was done using the log scale.

| Method | wave | | schrodinger | | convection | |
|---|---|---|---|---|---|---|
| | $\gamma_{\mathrm{critic}}$ | $C_{\mathrm{KL}}$ | $\gamma_{\mathrm{critic}}$ | $C_{\mathrm{KL}}$ | $\gamma_{\mathrm{critic}}$ | $C_{\mathrm{KL}}$ |
| FPCA | 3.1e-04 | 5.0e-03 | 3.0e-03 | 1.8e-02 | 7.6e-04 | 9.0e-03 |
| NN | 3.1e-03 | 3.4e-02 | 4.5e-03 | 9.9e-02 | 5.0e-03 | 5.4e-02 |
| Naive | 4.2e-03 | 3.1e-02 | 1.0e-03 | 2.9e-02 | 4.3e-03 | 2.7e-02 |
| Resolvent | 5.0e-03 | 1.7e-02 | 9.3e-04 | 1.2e-02 | 2.8e-04 | 7.1e-04 |

Table 1: TRPO common hyperparameters among methods. Selected critic learning rates, $\gamma_{\mathrm{critic}}$ and maximum Kullback Leibler divergences, $C_{\mathrm{KL}}$, for each method in each environment

Concerning the PPO algorithm, we used an entropy coefficient of 0.01, a weight for the critic loss function of 0.5 and clipped the gradient, so its norm is at max 0.5. We only tuned the learning rate, $\gamma_{\mathrm{PPO}}$. We present the selected hyperparameters in different environments in Table 2. The search for $\gamma_{\mathrm{PPO}}$ was done in the interval [1.0e-05, 1.0e-01], using a log scale.

| Method | wave | schrodinger | convection |
|--------|------|-------------|------------|
| FPCA | 5.6e-03 | 2.6e-03 | 5.0e-03 |
| NN | 7.6e-04 | 9.9e-04 | 1.7e-04 |
| Naive | 1.7e-02 | 3.3e-03 | 2.4e-02 |

Table 2: PPO common hyperparameters among methods. Selected learning rate $\gamma_{\text{PPO}}$, for each method in each environment

At last, the FPCA and resolvent methods have additional tunable hyperparameters. When using FPCA, we determine the number of eigenbasis used, by dropping those with those with small eigenvalues; concretely, we select as many components as necessary to explain a certain proportion of the observed variance. We tune for this percentage % Var. When using the resolvent method, we need to tune for the degree of penalization $p$ and a scalar $\alpha$. We present the selected hyperparameters in Table 3. The range of search for the $\alpha$ hyperparameter was the interval $[1.0e\text{-}03, 1.0e02]$, and the degree $p$ was searched in the set $\{1, 2, 3\}$, and the percentage of explained variance in TRPO was searched in the set $\{90\%, 95\%, 99\%, 99.9\%\}$.

| Environment | FPCA % Var | | Resolvent | |
|-------------|------|------|------|------|
| | PPO | TRPO | $\alpha$ | $p$ |
| wave | 99.9% | 99% | 2.5e-01 | 1 |
| schrodinger | 99% | 95% | 1.3e-2 | 1 |
| convection | 90% | 99.9% | 4.9e-2 | 2 |

Table 3: Additional hyperparameters selected for the FPCA method, using either PPO or TRPO and hyperparameters for the resolvent method, in each environment

## C  Appendix: Algorithmic Outline of Functional Linear Policy Optimization

The following algorithm outlines the practical implementation of our framework. To handle the functional nature of the states, we project the discrete spatial measurements onto the continuous functional basis space. The operator inversion unit corresponds to the designated operator inversion from sections 2.1 to 2.3.

**Require:** Functional basis $\Phi$ with $K$ basis functions, Fixed observation grid $\mathcal{G}$.
**Require:** Initial functional linear policy $\pi_\theta(a|x)$ and value function $V_\phi(x)$.
**Require:** Hyperparameters: Discount factor $\gamma$, GAE parameter $\lambda$, Number of Episodes $E$, Steps per episode $T$.

1: **for** episode $e = 1, \ldots, E$ **do**
2:   Get full observations: $\mathbf{o}_0$ from environment $\mathcal{E}$
3:   **for** step $t = 0, \ldots, T-1$ **do**
4:     Project regular measurements onto basis $\Phi$: $\mathbf{c}_t \leftarrow \text{get\_fd\_coefs}(\mathbf{o}_t, \mathcal{G})$     {*Representation Step*}
5:     Sample actions $\mathbf{a}_t \sim \pi_\theta(\cdot|\mathbf{c}_t)$ and evaluate log-probabilities $\log \pi_\theta(\mathbf{a}_t|\mathbf{c}_t)$     {*Action Selection*}
6:     Step environments: $\mathbf{o}_{t+1}, \mathbf{r}_t \leftarrow \mathcal{E}.\text{step}(\mathbf{a}_t)$
7:   **end for**
8: **end for**
9: Compute values $V_\phi(\mathbf{c}_t)$ for all stored states     {*GAE*}
10: **for** step $t = T-1$ down to $0$ **do**
11:   $\delta_t \leftarrow \mathbf{r}_t + \gamma V_\phi(\mathbf{c}_{t+1}) - V_\phi(\mathbf{c}_t)$
12:   $\mathbf{A}_t \leftarrow \delta_t + \gamma\lambda\mathbf{A}_{t+1}$   (with $\mathbf{A}_T = 0$)
13:   $\mathbf{R}_t \leftarrow \mathbf{A}_t + V_\phi(\mathbf{c}_t)$
14: **end for**
15: Apply designated operator inversion/penalization framework directly on $\mathbf{c}_t$     {*Operator Inversion*}
16: Update policy $\theta$ and value function $\phi$ using the projected functional states.     {*Policy Update*}

# D  Appendix: pairwise comparison of policies at the end of training

As a complement to Figures 1, 2, and 3, in this annex, we provide a thorough comparison of the model's final performance in different environments. These comparisons do not replace the figures; rather, they validate that we are indeed observing significant differences among methods instead of just noise.

To provide a rigorous statistical comparison among methods, we use non-parametric statistical tests: the final reward is not normally distributed and is bimodal for the NN policy when using TRPO. Concretely, for a given environment and algorithm (PPO or TRPO), we compare all the different methods, first using a Friedman test (Friedman, 1937), to see if there is any significant difference among methods and. And if there are, we proceed to make pairwise comparisons among all the methods using Wilcoxon's signed-rank test (Mann & Whitney, 1947). For completeness, we also provide the p-values obtained without supposing that the measurements are paired. For instance, we considered three methods when using the PPO algorithm, so three pairwise comparisons are done on the same data, and we considered five methods when using the TRPO algorithm, so ten comparisons are done on the same data. Because we are doing multiple comparisons on the same data, we correct the obtained p-values using the Benjamini-Hochberg procedure (Benjamini & Hochberg, 1995). We present the obtained results in Table 4.

| Method A | Method B | convection | | schrodinger | | wave | |
|---|---|---|---|---|---|---|---|
| | | Paired | Unpaired | Paired | Unpaired | Paired | Unpaired |
| PPO | | | | | | | |
| Naive | FPCA | 0.025 | 0.145 | 0.003 | 0.078 | 0.423 | 0.855 |
| NN | FPCA | $< 0.001$ | $< 0.001$ | $< 0.001$ | $< 0.001$ | $< 0.001$ | $< 0.001$ |
| NN | Naive | $< 0.001$ | $< 0.001$ | $< 0.001$ | $< 0.001$ | $< 0.001$ | $< 0.001$ |
| TRPO | | | | | | | |
| Naive | FPCA | 0.334 | 0.407 | $< 0.001$ | $< 0.001$ | 1.000 | 1.000 |
| NN | FPCA | $< 0.001$ | $< 0.001$ | $< 0.001$ | $< 0.001$ | $< 0.001$ | $< 0.001$ |
| NN | Naive | $< 0.001$ | $< 0.001$ | $< 0.001$ | $< 0.001$ | $< 0.001$ | $< 0.001$ |
| Resolvent | FPCA | $< 0.001$ | $< 0.001$ | $< 0.001$ | $< 0.001$ | 1.000 | 1.000 |
| Resolvent | Naive | $< 0.001$ | $< 0.001$ | 0.297 | 0.211 | 1.000 | 1.000 |
| Resolvent | NN | 0.014 | 0.407 | $< 0.001$ | $< 0.001$ | $< 0.001$ | $< 0.001$ |

Table 4: Pairwise comparison of final performances, using Wilcoxon's signed-rank test, for different methods, in three environments, using Benjamini-Hochberg correction for multiple testing.

Using the results from Table 4 and Figure 3, when using PPO, we observe that both functional methods outperform the NN policy in all environments. Likewise, the naive and FPCA functional methods significantly outperform the NN policy in all environments, and so does the resolvent method in the Schrödinger and wave environments. The resolvent method is significantly different from the NN policy in the convection environment when considering paired measurements but not when measurements are supposed to be unpaired. When comparing the FPCA and naive functional methods, we observe similar performances in the wave and convection environments and different performances in the Schrödinger environment.

# E  Additional experiments: FLPs on an irregular grid

Theoretically, FLPs depend on the state function rather than on the observed grid of points; the grid may be regular or irregular. The functional state *should* be resolution-invariant. However, the FDA literature also shows that increasing the number of points, improves the estimation of the functional coefficients used to represent the state functions (Ramsay & Silverman, 2005). In this section, we investigate the impact of the number of observed points (*i.e.* the grid size) $n_{\text{grid}} \in \{8, 9, 10, 11, 12\}$, on the performance of FLPs.

We do this by *randomly* sampling $n_{\text{grid}}$ observed points from the full grid and using these points to estimate the functional state $s(\cdot)$. Thus, training is performed on an irregular grid that *changes every step of the*

*episode.* Once the FLP has been trained, it is evaluated in the environment using the full grid with 12 points, *without retraining the FLP*. We then compare the learning trajectories and obtained performances for the FLP, depending on $n_{\mathbf{grid}}$ using 10 random seeds, and this is done for each one of our three PDE environments. FLPs are trained using the "naive" functional version of PPO. We reuse the same hyperparameters that were obtained by tuning the experiments from Section 3, we remind that those are presented in Table 2.

To be more specific, we use a Fourier basis with seven basis functions, and as with experiments from Section 3, both the constant $c$ and the functional parameter $\beta(\cdot)$ are initialized as 0. To obtain fair estimates of the learning trajectories and the performances of the learnt FLPs, we use 10 random seeds. For comparison, we also use the performances of the zero and LQR controllers.

We present the obtained learning trajectories in Figure 4. Similarly as in Figure 1, we show the two-sided 0.95-confidence interval for the median, calculated using bootstrap.

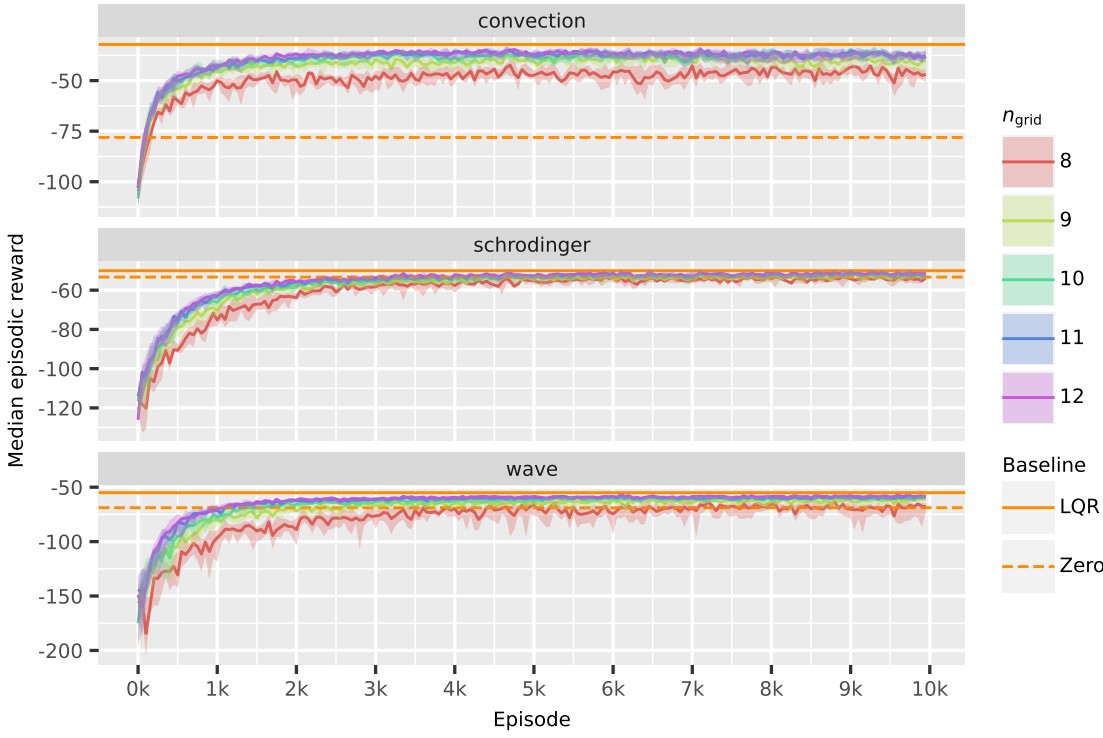

Figure 4: Median Episodic Reward vs. Episode for Different Number of Missing Measurements.

We observe that, in the three PDE environments, the FLP learning trajectories when there are 12, 11 or 10 measurements are similar, but when there are 8 measurements, performances decrease.

We confirm this by evaluating the trained FLPs on the environment with the full grid, *without retraining* the FLPs. For each random seed and for each $n_{\mathrm{grid}}$, using the trained FLP, we run 200 episodes and compute the median episodic reward. We repeat this process for the ten random seeds and visualize the final performance in Figure 5:

We can see that the FLPs, trained on an irregular grid of 11, 10 or 9 points seem to be efficient in an environment with a regular 12 point grid, without retraining. Indeed, the performances at the end of training in an irregular grid (Figure 4) are similar to the performances obtained when using the full grid (Figure 5). And we confirm that there is an important decrease in performance, when training of a 8 point grid. This could be due to the fact of using 8 points to estimate 7 Fourier functional coefficients, without adding a functional regularization.

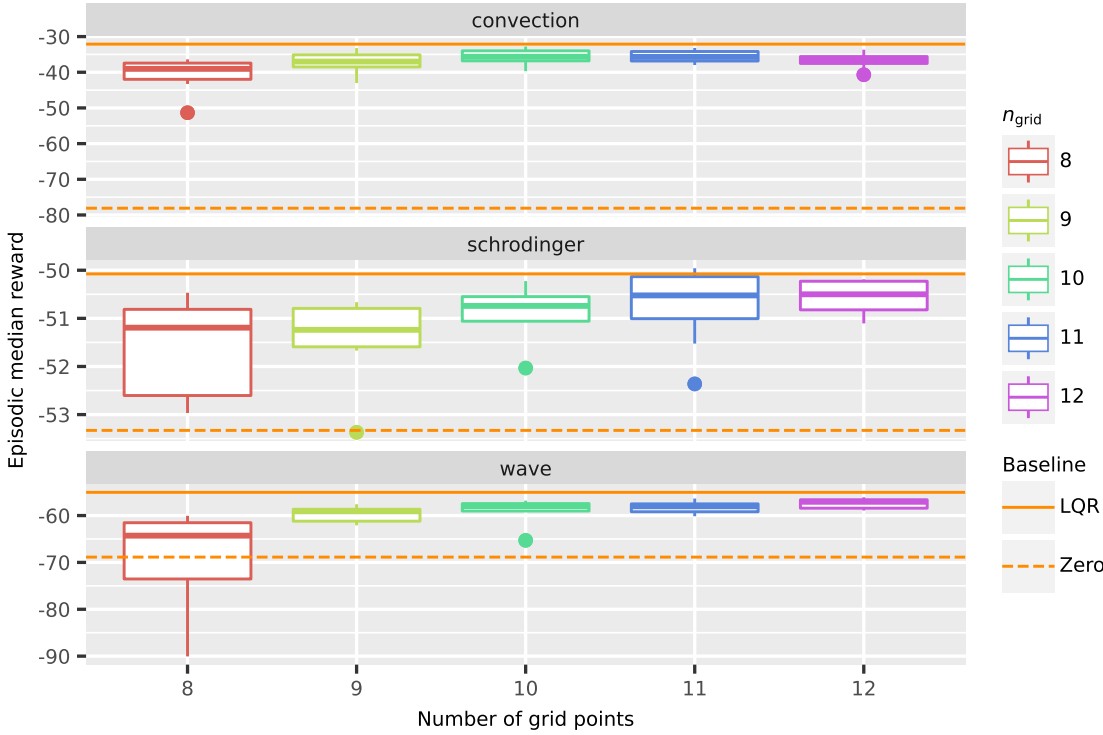

Figure 5: Median Episodic Reward of Trained FLPs on Environment without Missing Measurements

This points toward further work: in FDA, it is common to use a functional regularization to estimate the functional coefficients. In this paper, we did not do that, instead the functional coefficients are estimated by minimizing the MSE, without regularization. Moreover, using a B-spline basis may yield more robust results in this irregular grid setting. Using robust functional representations may improve the performance of FLPs when the grid is irregular.

## F   Appendix: Computational complexity

The main computational overhead specific to our functional approach lies in handling the infinite-dimensional operator inversion via Functional Data Analysis (FDA) techniques.

**Basis Size ($K$)**   : By projecting the functional states onto a choice of $K$ basis functions, the infinite-dimensional operator inversion reduces to the inversion of a $K \times K$ matrix. The computational cost of this step scales as $O(K^3)$. Since $K$ is typically chosen to be relatively small ($K \ll$ dimension of raw discretization), this is highly efficient.

**Discretization & Timestamps ($N_{\mathbf{obs}}$)**   : The number of discrete spatial measurements or timestamps per function only impacts the initial step—projecting the raw data onto the functional basis. This preprocessing step scales linearly with the number of discretization points, $O(N_{\mathrm{obs}})$, and is performed before evaluating the policy or computing gradients.

**Scaling**   : Consequently, once the functional representation is obtained, the RL training loop scales strictly with the complexity of the functional state (the basis size $K$) rather than the resolution of the physical grid, avoiding the curse of dimensionality.

