# OpenReview forum: "Trust Region Policy Optimization for Functional Linear Policies"
_TMLR — Decision pending for TMLR_

### Review · Reviewer_qym7 · 2026-05-07

**Summary Of Contributions:**

The authors introduce Functional Linear Policies (FLPs) to address Reinforcement Learning (RL) problems characterized by high-dimensional, continuous state spaces, such as spatial or temporal measurements in Partial Differential Equation (PDE) control tasks. Instead of flattening these measurements into a standard neural network, they adapt the Trust Region Policy Optimization (TRPO) and Proximal Policy Optimization (PPO) algorithms to operate directly on infinite-dimensional functional states. To overcome the resulting infinite-dimensional operator inversion problem during the TRPO update, the authors incorporate techniques from Functional Data Analysis (FDA), specifically finite basis projection (the "Naive" approach), Functional Principal Component Analysis (FPCA), and a resolvent approach. In three PDE control environments, the proposed FLPs demonstrate significantly more stable training and better final performance than standard Multi-Layer Perceptron (MLP) policies, achieving results close to an ideal LQR controller.

**Key Strengths:**
* **Theoretical Adaptability:** The mathematical formulation is sound. The authors successfully extend the core TRPO performance improvement bound to accommodate functional linear policies (Proposition 1).
* **Empirical Validation:** The experimental results convincingly demonstrate that integrating spatial and temporal priors through basis expansion (even in its simplest form) yields far more stable and robust control policies than the naive flattening of state observations into a vanilla MLP.

**Key Weaknesses:**
* **Misleading Novelty Claims and Missing Literature:** The paper strongly claims that functional linear models "have never been used in a RL context". This narrative is highly misleading as it entirely ignores vast bodies of closely related literature. Specifically, it overlooks classical Basis Function Approximation (e.g., Fourier basis features) which was a standard RL practice before Deep RL, Kernel RL methods that implicitly handle infinite-dimensional spaces via Reproducing Kernel Hilbert Spaces (RKHS), and modern Physics-Informed Machine Learning (PIML) or Neural Operators heavily used for continuous PDE control.
* **Discrepancy Between Narrative and Empirical Evidence:** The paper dedicates significant theoretical effort to introducing complex FDA techniques (FPCA and Resolvent estimators) to solve the infinite-dimensional operator inversion problem. However, the experiments reveal that the simplest "Naive" approach (finite basis projection) performs just as well as FPCA. This suggests the intrinsic dimensionality of the chosen test environments is actually quite low, making the sophisticated infinite-dimensional storytelling feel overblown and unnecessary for the tasks at hand.

**Audience:**

No

**Audience Explanation:**

I selected "No" not because the general topic lacks potential value, but because the paper's current narrative creates a "lose-lose" situation for TMLR's audience regarding genuine academic interest. The interest it might generate is either based on misunderstandings or negated by a lack of substantial novelty.

Specifically, the audience can be divided into two groups regarding this paper:

**1. Readers unfamiliar with classical RL or control theory:** These readers might genuinely find the paper "interesting" because it claims to introduce a groundbreaking, first-of-its-kind approach to infinite-dimensional reinforcement learning. However, this interest would be built on a misleading narrative and historical inaccuracies. Accepting the paper in its current form would propagate a distorted view of the field, making readers believe that functional analysis is a completely novel intervention here, while ignoring decades of identical underlying mathematical principles in Kernel RL and Basis Function Approximation. It is not in the audience's best interest to be misinformed.

**2. Informed readers (experts in RL, Control, or PIML):**
Readers who possess the background to not be misled will likely find the findings uninteresting. They will quickly recognize the "Functional Linear Policies" as a repackaging of standard basis expansion techniques. Furthermore, the core empirical finding—that the mathematically heavy FDA tools (FPCA/Resolvent) offer no significant advantage over the simplest baseline (the Naive Fourier basis)—undermines the entire theoretical buildup. For experts, realizing that the paper merely rediscovers old methods and applies them to environments where intrinsic dimensionality is low enough for naive methods to succeed, provides little to no new scientific insight.

**Conclusion:** Until the paper undergoes a major narrative rewrite to honestly position its true contribution—perhaps as a comparative empirical study of classical basis methods combined with modern TRPO, rather than a theoretical breakthrough—it does not provide scientifically valuable findings that would genuinely benefit the TMLR audience.

**Broader Impact Concerns:**

I do not foresee any direct negative ethical implications or societal harms arising from this work.
The paper focuses on theoretical and algorithmic advancements in Reinforcement Learning, specifically extending TRPO and PPO to functional linear policies for PDE control tasks.
While the authors mention potential downstream applications such as the personalized control of wearable devices, the core contribution remains methodological. The proposed techniques do not inherently raise significant concerns regarding bias, privacy, surveillance, or safety that would necessitate a mandatory Broader Impact Statement.
Therefore, I have no broader impact concerns regarding this submission.

**Claims And Evidence:**

No

**Claims Explanation:**

I selected "No" because there is a significant disconnect between the paper's strong narrative claims and both the historical context of the field and the provided empirical evidence.

**1. Misleading Claims of Novelty and Missing Literature**
The authors heavily claim that functional linear models and the handling of infinite-dimensional states "have never been used in a RL context." This claim is historically inaccurate and misleading. The core mathematical operation of the proposed "Naive approach" (projecting continuous states onto a finite Fourier basis) is conceptually identical to classical Basis Function Approximation, which has been deeply explored in RL. Furthermore, implicitly handling infinite-dimensional spaces without explicit operator inversion is a standard practice in Kernel RL (using RKHS).

The authors also ignore highly relevant modern literature that tackles infinite-dimensional RL and the use of Fourier features in continuous control. To make accurate claims, the authors must contextualize their work against papers such as:
* "Reinforcement Learning for Infinite-Dimensional Systems"
* "Functional Regularization for Reinforcement Learning via Learned Fourier Features"
* "Fourier Features in Reinforcement Learning with Neural Networks"

**2. Empirical Evidence Contradicts the Claimed Necessity of Complex FDA Tools**
The narrative dedicates substantial theoretical effort to introducing complex FDA techniques (FPCA and Resolvent estimators) to solve the "operator inversion in infinite-dimensional spaces" challenge. However, the empirical evidence presented in Figure 1 and Figure 2 shows that the simplest "Naive" approach performs just as well as the sophisticated FPCA method. This strongly suggests that the intrinsic dimensionality of the tested PDE control environments is relatively low. Therefore, the empirical evidence does not convincingly support the claim that these complex infinite-dimensional operator inversion techniques are practically necessary for the tasks at hand.

**Requested Changes:**

To secure a recommendation for acceptance, the paper requires a major rewrite of its narrative and literature review to ensure all claims are historically accurate and well-supported by the evidence.

**Critical Adjustments (Required for Acceptance):**

1. **Reframe the Novelty Claims:** The authors must explicitly remove statements claiming that functional linear models or infinite-dimensional states "have never been used in a RL context." The core contribution must be honestly repositioned as: "investigating the integration of classical functional basis reduction techniques (FDA) with modern policy optimization algorithms (TRPO/PPO) to achieve stable control in continuous PDE environments."

2. **Revise and Expand the Literature Review:** A dedicated section must be added to discuss the historical and mathematical connections between the proposed Functional Linear Policies (FLPs) and existing frameworks. The authors must cite and discuss:
   * *Classical Basis Function Approximation:* Acknowledge that the "Naive approach" is essentially classical Fourier basis function approximation, a standard technique in early RL.
   * *Kernel RL:* Discuss how FDA basis projection relates to the implicit handling of infinite-dimensional spaces via RKHS in Kernel RL methods.
   * *Recent Infinite-Dimensional RL & Fourier Features:* The authors must engage with recent works that tackle similar challenges in continuous/infinite-dimensional spaces, explicitly including:
     * `"Reinforcement Learning for Infinite-Dimensional Systems"` (to contrast system-level infinite dimensions with observation-level FDA).
     * `"Functional Regularization for Reinforcement Learning via Learned Fourier Features"` (to contrast FLPs with regularized learned features).
     * `"Fourier Features in Reinforcement Learning with Neural Networks"` (to contrast FLPs with input-preprocessing techniques).

3. **Reconcile Narrative with Empirical Evidence (Naive vs. FPCA):**
   The authors must explicitly address the empirical finding (shown in Figures 1 and 2, and Table 4) that the Naive method performs identically to or slightly better than the sophisticated FPCA method. The narrative must be adjusted to acknowledge that the tested controlgym environments likely possess a low intrinsic dimensionality, which renders the complex "operator inversion" machinery (FPCA/Resolvent) practically unnecessary for these specific tasks. The authors should moderate their claims about the necessity of these advanced FDA tools.

**Recommended Adjustments (To Strengthen the Work):**

1. **Discuss Computational Complexity:** Since FLPs avoid the heavy backpropagation of Deep Neural Networks and rely on finite basis projections, a theoretical or empirical comparison of the computational cost (e.g., FLOPs or wall-clock inference time) between FLPs and the NN baseline would strongly highlight the practical benefits of this method, especially for resource-constrained applications.

2. **Clarify the Positioning Against Neural Operators:** While empirical comparison might be beyond the scope of a revision, the authors would significantly strengthen the paper by briefly discussing in the introduction or conclusion why one might choose an FLP over modern Physics-Informed Machine Learning (PIML) architectures like Neural Operators (e.g., FNO) for PDE control (e.g., emphasizing strict interpretability or lower sample complexity).

---

> ### Author Response · Authors · 2026-06-09
>
> We thank the reviewer for this constructive review; we agree that the first version of the article lacked important references from the literature and some claims were imprecisely worded; we have substantially revised the related work and main contributions paragraphs accordingly (see below). **However, there appears to be some confusion between classical value function approximation from the RL literature and what we propose in this paper**; we acknowledge that this confusion may have originated from our initial lack of literature review and believe it is at the root of the reviewer's concern about novelty. We address this directly.
>
> Using a finite functional basis (such as Fourier) was indeed one of the earliest methods used for value approximation: for a finite-dimensional state vector $s\in \mathbb{R}^p$, we have $V(s) \simeq \sum_k \omega_k \phi_k(s)$, with learnable weights $(\omega_k)$. These weights can be learned by minimizing the Bellman error. In contrast, we use a finite functional basis to express the state, which is a function, as a finite sum of functions $s(\cdot) = \sum_k a_k \phi_k(\cdot)$, expressing the state as an element of a Hilbert space. Unlike early RL value approximation methods, representing the functional state this way does not impose anything on the form of the state value function $V(\cdot)$. Our contribution is a principled policy-based method that operates directly on these functional coefficients $(a_k)_k$, not a value-based method approximating the $Q$ or $V$ functions.
>
> An analogy may help clarify the distinction. Generalized Additive Models (GAMs) and Functional Data Analysis (FDA) can both employ a Fourier basis, yet they are fundamentally different: in a GAM, the basis functions are applied to scalar covariates to model nonlinearity in the regression function; in FDA, the basis represents the *object of study itself* as an element of a function space. Concluding that our method is equivalent to classical RL basis function approximation because both use a Fourier basis is analogous to concluding that GAMs and FDA are the same method.
>
> Now that this misunderstanding has been clarified, we are working on a second version of the paper incorporating the requested changes.
>
> - **Reframe novelty claims:** We reworded our claims, making them more specific and reframing the numerical result section as "investigating the integration of classical functional basis reduction techniques (FDA) with modern policy optimization algorithms (TRPO/PPO) to achieve stable control in continuous PDE environments."
>
> - **Revise and Expand the Literature Review:** We included new paragraphs with relevant works, allowing us to better position our contributions. We include below the aforementioned review, which would replace the last two paragraphs of the original Introduction.
>
> - **Reconcile narrative with empirical evidence:** We added explicitly that we empirically find that the naive and more intricate FDA methods (FPCA and resolvent approaches) yield similar results, at the end of Section 3.3 and in Section 4.
>
> We include below (in a second comment) the revised paragraphs, which would replace the last two paragraphs of the original Introduction.

---

> > ### Author Response · Authors · 2026-06-09
> > **The following 4 paragraphs would replace the last 2 paragraphs of the Introduction section**
> >
> > **Related work:** This functional setting has received increasing attention. Notably, functional states naturally arise in control problems where system evolution is governed by Partial Differential Equations (PDE). Farahmand et al. (2016) proposed an RL approach for PDE control using a value-based method: regularized fitted Q-iteration, adapting Q-learning for handling infinite dimensionality via a reproducing kernel Hilbert space representation. Later, Pan et al. (2018) addressed control problems with function-valued actions by exploiting spatial regularity in the action space through "action descriptors", reducing the infinite-dimensional problem to a finite-dimensional one, and then proposing a deterministic policy gradient algorithm in that context to improve actor and critic convolutional networks. At last, this infinite-dimensional setting may arise from "large-scale RL", where an infinite number of agents must learn how to act; Zhang & Li (2025) tackle this setting by proposing an adapted RKHS approach.
> >
> > An alternative to using RKHS or CNNs is to use a finite functional basis (Fourier, splines, etc.) expansion for function value approximation. This was one of the earliest forms of function approximation techniques for continuous multivariate states; early examples include the work of Schweitzer & Seidmann (1985) or Bertsekas & Tsitsiklis (1996, Chapter 3), approximating the state or the state-action value functions as a linear combination of non-linear features of the state. For instance, Konidaris et al. (2011) approximate the state value function as $\bar{V}(s)\simeq\sum_{k=1}^m \omega_k \phi_k(s)$, with learnable weights and $\{\phi_k\}$ a multivariate Fourier basis. More recently, this approach was adapted to allow for non-linear function approximation with the work of Brellmann et al. (2023) and Li & Pathak (2021). Both works propose parameterizing the Q-function with a neural network using a multivariate Fourier transform for the state space, which is then fed into an MLP to approximate the state-value function. The former work uses a multivariate Fourier transform, while the latter incorporates a learned multivariate Fourier transform. Additionally, this functional setting is also present in image-based RL; for instance, Li & Pathak (2021) propose a Q-learning method incorporating a learnable Fourier embedding into the Q-network, improving sample efficiency for image-RL problems.
> >
> > **Main contributions:** In contrast with existing methods from the literature, we do not use an RKHS approach nor function value approximation; instead, we propose a linear policy-based method, allowing us to tackle the scenario where the state is itself a function. In concrete terms, we introduce a family of policies, Functional Linear Policies (FLPs), which take continuous actions depending on functional states. These policies are direct adaptations of functional linear models (Cardot et al., 1999). Additionally, we prove that the main theoretical result from TRPO still holds for FLPs, and we adapt TRPO to propose practical algorithms to improve these policies. Prior work proposed the TRPO update in a classical RL context, with finite action and state spaces (Schulman et al., 2015). Our main theoretical result follows the same proof of Schulman et al. (2015, Theorem 1) and the proposed practical algorithms deal with issues arising from the highly dimensional setting by relying on classical FDA techniques: finite basis projection, FPCA (Wang et al., 2016) and a resolvent approach (Kreyszig, 2007). Additionally, based on these FDA techniques, we propose a Proximal Policy Optimization (PPO) update. At last, we investigate the integration of classical functional basis reduction techniques (FDA) with modern policy optimization algorithms (TRPO/PPO) to achieve stable control in continuous linear PDE environments.
> >
> > We would like to stress that, similar to previous work, we also use a finite basis expansion, but our use is fundamentally different from the aforementioned works, as we focus on the *policy* side. In the prior literature, the state is a finite-dimensional vector $s \in\mathbb{R}^d$, and the basis functions $\phi_k:\mathbb{R}^d \to \mathbb{R}$ are nonlinear features *of* the state, used to approximate the value function: $V(s) \approx \sum_k \omega_k \phi_k(s)$. In our framework, the state itself is a function $s(\cdot)\in L^2([0,1])$, and writing $s(\cdot)=\sum_k a_k\phi_k(\cdot)$ expresses the state as an element of a Hilbert space; the coefficients $a_k$ are its coordinates. We then define the *policy* directly over these coordinates. Note that in our experiments our critic does take these coordinates as inputs to a neural network, similarly to prior work (Konidaris et al., 2011); however, this is not our contribution. Our contribution is the functional parameterization of the policy itself, which is what enables the Hilbert-space geometry underlying Proposition 1.

---

> > > ### Comment · Reviewer_qym7 · 2026-06-14
> > >
> > > Thank you for the detailed response and for providing the revised text for the Introduction and Related Work sections.
> > > I appreciate the analogy between GAM and FDA provided in your response. This successfully clarifies the distinction between classical RL basis function approximation (used for feature engineering in value functions) and your proposed method (defining a policy over the Hilbert-space coordinates of the functional state).
> > >
> > > I agree that this forms a valid theoretical distinction. Furthermore, your incorporation of the RKHS and Fourier literature, alongside the softened novelty claims and the explicit acknowledgment of the 'Naive' baseline's performance, improves the paper's contextualization.
> > >
> > > **However, two critical points from my previous feedback remain unaddressed in your rebuttal and the proposed revised text:**
> > >
> > > **1. Positioning Against Physics-Informed ML (PIML) and Neural Operators:**
> > > While you contrasted your method with RKHS and CNNs, your revised text still ignores a dominant modern paradigm for infinite-dimensional PDE control: Neural Operators (e.g., Fourier Neural Operator [FNO], DeepONet). Since your environments are PDE control tasks, it is necessary to explicitly mention this family of methods in your Related Work. You do not necessarily need to run experiments against them, but you should theoretically position your Functional Linear Policies against Neural Operators (e.g., highlighting that FLPs offer an interpretable, non-NN, lightweight alternative that focuses purely on policy parameterization).
> > >
> > > **2. Theoretical Motivation for the Non-NN Paradigm (Why do NNs fail here?):**
> > > Your revised *Main Contributions* paragraph states that you propose a linear policy instead of an NN. However, the manuscript still does not rigorously explain *why* the standard MLP baseline fails so catastrophically in your experiments. To make your non-NN functional route compelling, you must add a brief discussion analyzing the failure modes of naive MLPs on spatial-temporal PDE measurements. Is it due to a lack of spatial inductive biases (smoothness priors)? Spectral bias? Susceptibility to measurement noise? A formal discussion giving the readers insight into *why* the functional FDA route prevents the instability seen in MLPs is essential to complete your paper's motivation.
> > >
> > > Once these two final points (Neural Operators literature and a formal discussion on NN instability) are **satisfactorily addressed and integrated** into the manuscript's text, **I would be inclined to recommend acceptance.**

---

> > > > ### Author Response · Authors · 2026-06-18
> > > >
> > > > We thank the reviewer for their feedback.
> > > >
> > > > We added into the updated related work section a paragraph mentioning neural operator approaches, these are indeed relevant as they have successfully been used to simulate PDEs and even have been used for PDE control tasks. As the reviewer points out, our approach offers a lightweight, interpretable alternative, which we have mentioned in the new version of our paper.
> > > >
> > > > Concerning *“Why NNs fail here”*, we cannot know for sure, there are multiple possible reasons. In our experiments, the agent is intentionally provided with a limited number of spatial observations (sensors). Under this sparse-data regime, standard MLPs face two intuitive hurdles that our functional approach resolves:
> > > >
> > > > **Lack of spatial bias:** As pointed out by the reviewer MLP treats sparse sensor measurements as isolated, independent inputs. It possesses no inherent understanding of the spatial continuity between spatial grid sensors. In contrast, our approach projects these sparse points onto a continuous functional basis, effectively reconstructing the global physical state. The policy reacts to the smooth global geometry rather than disjointed local coordinates. Positional encoding, recurrent networks could be alternative but possibly with less sample efficiency (see next point).
> > > >
> > > > **Sample Efficiency and complexity:** Learning physical control using “only” few thousands episodes requires high sample efficiency. By encoding the spatial inductive bias directly into a small number of basis functions, our functional policy scales strictly with the complexity of the underlying state (see Annex F), resulting in a significantly compressed parameter space that is much easier to optimize over fewer iterations. The MLP may learn a good control but it may need much more episodes.
> > > >
> > > > We have updated the Limitations section to highlight these practical, intuition-driven differences.

---

> > > > > ### Comment · Reviewer_qym7 · 2026-06-19
> > > > >
> > > > > Thanks for the revision. I have checked the updated manuscript and confirmed that my main concerns have been addressed.

---

### Review · Reviewer_D65n · 2026-05-21

**Summary Of Contributions:**

This paper integrates Functional Data Analysis (FDA) with reinforcement learning to handle infinite-dimensional functional states. The authors also introduce Functional Linear Policies (FLPs). Theoretically, a major theoretical contribution is extending the monotonic policy improvement bounds of TRPO to bounded infinite-dimensional functional Borel spaces. They also extend PPO to FLPs and evaluate their methods on three PDE control tasks.

Key Strengths:
1. Proposition 1 successfully generalizes TRPO's mathematical convergence bounds to infinite-dimensional functional Borel spaces.
2. This work formulates clear mathematical workarounds (FPCA, resolvents) to address non-invertible expected information operators in $L^2$ spaces.

Key Weaknesses:
1. While the theory addresses infinite-dimensional spaces, the actual implementation relies on basis truncation or FPCA projection. This maps continuous functions back into finite coordinate vectors. In the latter senario, it seems standard finite-dimensional TRPO/PPO would just do the work.
2. Proposition 1 is basically a very natural extension from the original TRPO work.

**Audience:**

Yes

**Audience Explanation:**

Yes. Researchers working at the intersection of deep reinforcement learning and continuous control will probably find the mathematical adaption of TRPO to $L^2$ spaces and the operator inversion workarounds relevant.

**Claims And Evidence:**

No

**Claims Explanation:**

1. The major concern is that the empirical evaluation fails to validate the claimed advantages of functional reinforcement learning due to critical gaps in the numerical experiments. The primary motivation for treating states as continuous functions is to handle varying sensor granularities or irregular meshes. However, all experiments use a fixed spatial grid. As mentioned, it seems standard finite-dimensional TRPO/PPO would just do the work.

**Requested Changes:**

Critical Changes (Required for Acceptance):

1. It is emphasized a lot that the core motivation of introducing Functional Data Analysis (FDA) to RL is to achieve resolution-invariant or mesh-independent control. The authors must provide empirical evidence demonstrating that the proposed functional policies can generalize across different discretization granularities (e.g., training on a lower-resolution or irregular grid and testing on a higher-resolution grid without retraining). Otherwise, it is obscure why this extension is practically different from the traditional TRPO/PPO....

2. Since handling infinite-dimensional functional operators involves iterative estimation (such as updating the covariance operator and eigenbasis in FPCA), the authors must include a comprehensive tracking of computational complexity. For example, this should feature wall-clock time comparisons and training curves against standard (finite) TRPO/PPO.

---

> ### Author Response · Authors · 2026-06-18
>
> We thank the reviewer for their constructive review. To address their concerns, we ran additional experiments studying the impact of an irregular grid on FLPs. We address the two requested changes below:
>
> **Irregular grid:** theoretically, FLPs depend on the state function, not on the observed grid of points; the grid may be regular or irregular, and the functional state should indeed be resolution-invariant. But the FDA literature also shows that the more points we have, the better the estimation of the functional coefficients used to represent the state functions. We investigate the impact of the size of the grid $n_{grid}\in \{8,9,10,11,12\}$ on the performance of FLPs. This was done by randomly sampling $n_{grid}$ observed points from the full grid, which we used to estimate the functional state. Thus, as suggested by the reviewer, we train using an irregular grid, that **changes every step of the episode**, and once the FLP is trained, we evaluate it in the environment using the full grid with 12 points, **without retraining the FLP**. We then proceed to compare the learning trajectories and obtained performances for the FLP, depending on $n_{grid}$ using 10 random seeds, and this is done for each one of our three PDE environments. We obtain similar performances when using 1 or 2 missing values and lower performances when using 3 or 4 missing values, which is coherent with the FDA literature. Thus, in this new version of the paper, we provide empirical evidence demonstrating that FLPs generalize across different granularities, but their performance does depend on the number of observed points: the more, the better. This also points toward further work: in FDA, it is common to use a functional regularization to estimate the functional coefficients; in the paper, we did not do that; instead, we simply minimized the MSE without regularization. Adding a functional regularization may yield robust estimates for the functional coefficients and improve the performance of FLPs when the grid is irregular. But we judge that studying this is beyond the scope of this paper. These results are presented in the Appendix E.
>
> **Computational complexity:** we added in the Appendix F a comprehensive tracking of the complexity of FLP, both in training and in inference.

---

### Review · Reviewer_ig2e · 2026-05-28

**Summary Of Contributions:**

The manuscript introduces functional linear policies (FLP), which define continuous control policies acting on functional observations of the state. FLPs are based on classical functional linear models, allowing an action to depend linearly on the observed function of the state. The authors show that the key theoretical guarantee of Trust Region Policy Optimization (TRPO) method based on relative performance identity and objective lower bound still holds: specifically, they adapt the classical TRPO performance bound to infinite-dimensional setting for FLPs. The paper extends TRPO and proximal policy optimization (PPO) method to optimize FLPs by leveraging tools from functional data analysis (finite-basis projection, functional PCA, and a resolvent method) to handle the high-dimensional function space. Finally, the authors evaluate their methods on three partial-differential-equation control tasks from the controlgym benchmark and compare FLP TRPO and PPO to baseline controllers: constant zero-action baseline and Linear-Quadratic Regulator (LQR). The results demonstrate that learning FLPs with the proposed updates yields more stable training and significantly higher rewards than a zero-action baseline, and outperforms a well-tuned standard neural-network policy in these environments.

**Audience:**

Yes

**Audience Explanation:**

This work lies at the intersection of reinforcement learning theory and applications involving high-dimensional (functional) state spaces. Researchers interested in RL with continuous or functional observations would find this approach sufficiently novel. The experimental results on PDE control tasks demonstrate practical relevance, especially to those working on control of physical systems or continuous spaces.

**Broader Impact Concerns:**

No broader impact concern has been found.

**Claims And Evidence:**

Yes

**Claims Explanation:**

The authors make clear, concrete claims regarding their policy class and learning algorithm, and these claims are generally well supported. The theoretical claim that the TRPO improved bound extends to FLPs is backed by a formal proposition in the appendix, the proof carefully follows established lemmas to ensure validity under the stated assumptions. The boundedness assumption on the functional state is noted and seems reasonable for the finite PDE domains considered. On the empirical side, the experiments are designed rigorously. The authors tune each method fairly, run several trials with shared random seeds, and report paired bootstrap confidence intervals. They use the same generalized advantage estimator for all methods. In every environment, both FLP TRPO and FLP PPO achieve median episodic rewards that are significantly better than the zero-action baseline, and in most cases exceed the performance of the neural-network controller, sometimes insignificantly. These illlustrated results convincingly support the claim that functional policies can improve performance and stability.

**Requested Changes:**

Correct the reward definition on page 7: the reward  based on non-negative loss should be non-positive.

The description of the practical learning algorithm could be clearer. Including pseudocode or a more detailed algorithmic outline would improve clarity.

The experiments focus on linear PDEs. The authors should comment on how their approach might extend to nonlinear dynamics, or explicitly note that the method is currently validated only for linear problems.

It would also be useful to discuss any computational costs (e.g. basis size, discretization of the state space, timestamps) and how performance scales with the complexity of the functional state.

---

> ### Author Response · Authors · 2026-06-12
>
> We thank the reviewer for their constructive feedback and valuable suggestions. Below we address each point individually and outline the changes made to the manuscript.
>
> **1. Reward Definition Correction**
>
> We have corrected the sign in the reward definition on page 7 of the revised manuscript.
>
> **2. Clarity of the Practical Learning Algorithm (Pseudocode)**
>
> We have added a pseudocode section in the Appendix detailing the practical implementation of both the Functional TRPO and Functional PPO algorithms. This outline explicitly shows the integration of the functional basis projection, the calculation of the functional policy gradient, and the regularized operator inversion step.
>
> *Note:* As mentioned in the text, our full implementation code remains publicly available.
>
> **3. Extension to Nonlinear Dynamics**
>
> The reviewer raises an important point regarding the scope of our validation. We focused our experiments on linear PDEs (such as the heat and convection-diffusion-reaction equations) because they admit well-established classical baselines (e.g., zero controllers, optimal linear quadratic regulators, or analytical linear controllers). This allowed us to benchmark our reinforcement learning agent against reliable ground truths.
>
> We have explicitly clarified in the Limitations section that our current empirical evaluation is validated on linear systems. However, we have also added a brief discussion on how the method extends to nonlinear dynamics: while the policy remains a functional linear model (acting as a first-order functional approximation), it can still be applied to nonlinear systems in a manner analogous to how standard linear policies are used in classical RL for complex, nonlinear continuous control tasks (e.g., robotics). For highly complex nonlinear states, future work will explore functional neural networks.
>
> **4. Computational Cost and Scaling Analysis**
>
> The main computational overhead specific to our functional approach lies in handling the infinite-dimensional operator inversion via Functional Data Analysis (FDA) techniques. We have added a dedicated paragraph in the Annex detailing this computational complexity analysis to clarify how our method scales.
>
> - *Basis Size (K)*: By projecting the functional states onto a choice of K basis functions (e.g., B-splines or Fourier bases), the infinite-dimensional operator inversion reduces to the inversion of a K×K matrix. The computational cost of this step scales as O(K^3). Since K is typically chosen to be relatively small (K≪dimension of raw discretization), this is highly efficient.
>
> - *Discretization & Timestamps (N)*: The number of discrete spatial measurements or timestamps per function only impacts the initial step—projecting the raw data onto the functional basis. This preprocessing step scales linearly with the number of discretization points, O(N), and is performed before evaluating the policy or computing gradients.
>
> - *Scaling*: Consequently, once the functional representation is obtained, the RL training loop scales strictly with the complexity of the functional state (the basis size K) rather than the resolution of the physical grid, avoiding the curse of dimensionality.

---

### Author Response · Authors · 2026-06-18
**General answer for reviews**

We want to thank the reviewers for their constructive feedback. We believe that answering their concerns allowed us to improve our paper. Likewise, we provided detailed answers to each reviewer. We have added a new version of the paper, a new appendix, and updated code to reproduce the extra numerical experiments. We have uploaded a revised version of the paper in OpenReview, which incorporates all the changes described below:


- Added new numerical experiments, evaluating FLPs when there is an irregular grid
- Reworded our claims, softening them and being more specific with our contributions
- Explicitly concluded that, in our numerical experiments, the naïve approach yields similar performances to more complex FDA approaches (resolvent, FPCA)
- Expanded our literature review, better positioning our contributions and explaining how they differ from other approaches
- Added in the Appendix F a comprehensive tracking of the complexity of FLPs, both in training and in inference
- Added in the Appendix C pseudocode to better present FLPs training.
- Stressed that we tested only linear PDEs, while also pointing out that FLPs could be applied to non-linear PDE control.
- Added to the discussion possible reasons on why using MLPs does not seem to work well
- Corrected the reward definition in Section 3.1